# Human brains construct individualized global rankings from identical few-shot learning input

Dongning Liu[1,2,3,4☯], Muzhi Wang[1,2,3,4,5☯*], Huan Luo[1,2,3,4*]

1 School of Psychological and Cognitive Sciences, Peking University, Beijing, China, 2 PKU-IDG/McGovern Institute for Brain Research, Peking University, Beijing, China, 3 Beijing Key Laboratory of Behavior and Mental Health, Peking University, Beijing, China, 4 Key Laboratory of Machine Perception (Ministry of Education), Peking University, Beijing, China, 5 Applied Computational Psychiatry Lab, Max Planck UCL Centre for Computational Psychiatry and Ageing Research, Queen Square Institute of Neurology and Mental Health Neuroscience Department, Division of Psychiatry, UCL, London, United Kingdom

☯ These authors contributed equally to this work.
* wangmuzhi@pku.edu.cn (MW); huan.luo@pku.edu.cn (HL)

## Abstract

Ranking—a ubiquitous relational structure—enables humans to organize complex information and overcome cognitive load, yet in real-world settings it is often inferred from sparse, few-shot learning of local pairwise relationships. How the human brain performs relational inference under such limited evidence remains unknown. We hypothesized that under few-shot learning, relational inference is shaped by inductive biases, such that individuals actively impose structured global relationships—often idiosyncratic—to constrain and unify limited local information. In a preregistered behavioral study combined with magnetoencephalography (MEG) recordings, we show that even after identical few-shot local pair learning, individuals construct stable and self-consistent, yet idiosyncratic, global rankings that diverge from the ground-truth order—a phenomenon not readily explained by classical computational models of transitive inference. MEG recordings further reveal that frontoparietal neural representations are reorganized to reflect each individual's subjective ranking rather than those of others. Together, these findings highlight the constructive and generative nature of human cognition: under sparse samples and limited computational resources, the human brain actively infers and imposes relational structure.

## Introduction

Humans possess a remarkable ability to rapidly infer generalizable rules from only a few examples and to apply these rules to novel inputs, a capacity known as few-shot learning [1,2]. This ability allows people to acquire new concepts or master new tasks with minimal experience. For example, the classic Wug Test shows that a child can correctly apply a novel label ("wug") to new objects after exposure to only a few exemplars [3]. Such efficiency is widely attributed to inductive bias—internal

**Data availability statement:** All data files are available at https://osf.io/gya95/.

**Funding:** This work was supported by the National Science and Technology Innovation STI2030-Major Project (2021ZD0204103; https://www.most.gov.cn/), the Science Fund for Creative Research Groups of the National Natural Science Foundation of China (T2421004; https://www.nsfc.gov.cn/), the National Natural Science Foundation of China (32541013; https://www.nsfc.gov.cn/), and the Fundamental and Interdisciplinary Disciplines Breakthrough Plan of the Ministry of Education of China (JYB2025XDXM504; http://www.moe.gov.cn/) to H.L. The funders had no role in study design, data collection and analysis, decision to publish, or preparation of the manuscript.

**Competing interests:** I have read the journal's policy and the authors of this manuscript have the following competing interests: H.L. is currently an academic editor of PLoS Biology.

**Abbreviations:** MEG, magnetoencephalography; RDMs, representational dissimilarity matrices; RSA, representational similarity analysis.

constraints that shape and restrict the learner's hypothesis space [4]. Rather than learning solely from surface-level samples, humans actively infer and construct internal representations of underlying relational structure to make sense of incoming information.

Inductive bias is fundamentally relational: instead of treating observations as isolated events, humans tend to project causal or structural relationships onto their environment [4–6]. One ubiquitous relational structure is ranking (e.g., rankings of tennis players, soccer teams, or universities), which provides an efficient means of organizing information and guiding decision-making [7–9]. Although a complete ranking formally requires either scalar values for each item or exhaustive pairwise comparisons, real-world learning typically relies on sparse samples of local pairwise relations [1,2,10–12]. For instance, given match outcomes for only a subset of player pairs, people can infer a global ordering that includes pairs that have never directly competed.

Because ranking involves stochastic relational inference under uncertainty [11,13], we hypothesized that inductive biases are engaged during few-shot learning of local pairwise relations. Under sparse evidence and limited computational resources [14,15], relational inference is therefore not a passive accumulation of evidence but is shaped by the active construction of a global ranking structure within each individual—an account we refer to as the *constructive ranking account* (Fig 1A, right). According to this account, individuals actively construct a unified and idiosyncratic global ranking from few-shot experience, which serves as an inductive bias to constrain learning of local pairs. As a result, even when exposed to identical local pairwise information, individuals tend to form stable and self-consistent, yet idiosyncratic, global rankings that deviate from the ground truth. This account contrasts sharply with the classical *independent-value account* (Fig 1A, left), previously used to explain transitive inference [16–19]. Under this account, each item is represented and updated independently as a point on a latent value axis, with no constraints imposed by a global ranking structure.

A classical framework for studying relational learning is transitive inference [20–26], in which relationships between non-adjacent items (e.g., A < C) are inferred from relationships between adjacent items (e.g., A < B, B < C). A related phenomenon, list linking [1,27,28], occurs when two independently learned sequences are rapidly integrated into a single ordered list after learning a single bridging relation. Numerous computational and neural network models have been proposed to account for such relational learning [1,16–18,29–33]. Here, we adapted the transitive inference paradigm within a few-shot learning framework to directly examine the role of inductive bias in relational inference.

We tested the constructive ranking hypothesis in a preregistered behavioral study and conducted magnetoencephalography (MEG) recordings in an independent cohort. We predicted that, after briefly learning a limited set of local pairs, participants would construct stable, self-consistent, and idiosyncratic global rankings. Behavioral experiments in two independent cohorts, as well as the behavioral component of the MEG experiment, confirmed these predictions. MEG recordings further showed

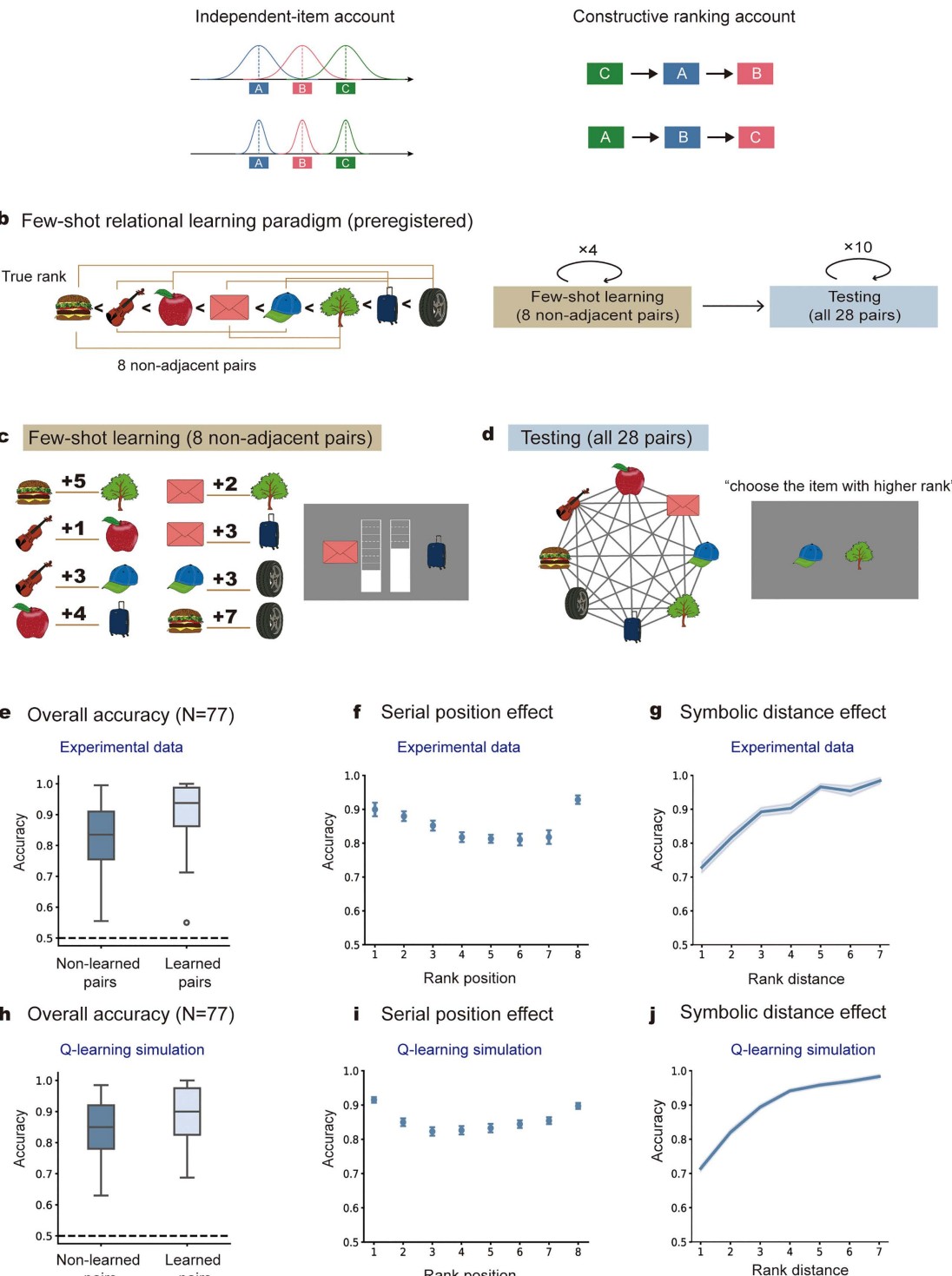

**Fig 1. Preregistered few-shot learning paradigm. (a)** Schematic illustration of two theoretical accounts. *Left:* The classical *independent-value account*, whereby local pairs are learned independently and evidence is passively accumulated toward the ground truth. *Right:* The *constructive ranking account*, where individuals actively construct global structure that constrains and unifies local pair learning. **(b)** *Left*: predefined artificial ranking (true

rank) of eight images (left-to-right corresponding to low-to-high ranks), unknown to subjects. Eight non-adjacent local pairs (connecting lines) were selected for few-shot learning (identical pairs for all subjects). *Right*: experiment consists of two phases: few-shot learning phase (brown box; each of the 8 pairs presented once per block across four blocks) and testing phase (blue box; each of the 28 pairs tested 10 times). Notably, subjects learned and were tested only on pairwise relationships, without explicit instruction to construct a full ranking of the eight images. **(c)** Few-shot learning phase. *Left*: eight non-adjacent pairs and their score distance (indicated by the number); *Right*: sample trial display, i.e., "Envelope" scores three units lower than "Suitcase". **(d)** Testing phase. *Left*: all 28 image pairs (connecting lines) were tested without feedback. *Right*: sample trial display, i.e., selecting the item with a higher score. **(e)** Grand averaged ranking accuracy in human participants, for learned pairs (dark blue; 8 pairs in few-shot learning) and non-learned pairs (light blue; remaining 20 pairs not directly learned) in human participants. **(f)** Grand averaged accuracy as function of rank position (serial position effect) in human participants. **(g)** Grand averaged accuracy as function of ranking distance (distance effect) in human participants. **(h)** Q-learning simulation results corresponding to e. **(i)** Q-learning simulation results corresponding to f. **(j)** Q-learning simulation results corresponding to g. The data underlying this Figure can be found in https://osf.io/gya95/.

that, following few-shot learning, neural similarity between items aligned with each participant's subjective ranking rather than with those of others. Together, these findings demonstrate that inductive biases shape few-shot relational learning, highlighting that under sparse samples and limited computational resources, the human brain actively infers and imposes global relational structure rather than passively accumulating information.

## Results

### Preregistered few-shot learning paradigm

We developed a preregistered behavioral paradigm (http://osf.io/dpxq4) to investigate how humans infer rankings of items (Fig 1B, left; left-to-right corresponds to low-to-high ranks) from brief learning of a limited set of pairs (8 out of 28 pairs; Fig 1B, left, connecting lines). The experiment comprises two phases (Fig 1B; right): few-shot learning phase (brown box) and testing phase (blue box). Participants first learned the relationship of eight image pairs (Fig 1C; e.g., "Suitcase" scores three units higher than "Envelope"), with each pair presented only once per block across four blocks. In the testing phase, participants judged all 28 possible image pairs without feedback, selecting the image they believed had a higher score (Fig 1D). Notably, subjects learned and were tested only on pairwise relationships, without explicit instruction to construct a full ranking of the eight items. Most crucially, all subjects were trained on the identical eight pre-determined local pairs (connecting lines in Fig 1B), ensuring that inter-subject differences in global ranking (as hypothesized in the preregistration) could not be simply attributed to different learning sets.

Notably, our preregistered study specified a target sample size of $N = 40$. After confirming the preregistered hypotheses in the first cohort ($N = 40$), we recruited an independent replication cohort ($N = 40$; three participants excluded due to below-chance accuracy). Because both cohorts exhibited consistent patterns in all preregistered analyses, we report combined results below ($N = 77$; see S1 and S2 Figs for cohort-specific results).

Overall, participants demonstrated good ranking performance after few-shot learning. As expected, directly learned image pairs (Fig 1E, light blue) were judged with high accuracy, confirming successful learning. Critically, participants also performed well on pairs that were never directly experienced (Fig 1E, dark blue), indicating robust transitive inference from few-shot exposure. Importantly, the inferred rankings exhibited hallmark signatures of ordinal ranking, including the "serial position effect"—higher accuracy for items at the beginning and end of the sequence (Fig 1F; repeated measure ANOVA, $F(7, 532) = 11.87$, $p < 0.001$)—and the "symbolic distance effect"—lower accuracy for item pairs with smaller rank differences (Fig 1G; linear regression slope $= 0.04$, $t(76) = 15.074$, $p < 0.001$), for both directly learned and indirectly inferred pairs (S3 Fig).

To further validate these canonical effects, we fit a standard Q-learning model [16,17] to participants' trial-by-trial choices (see Materials and methods). As shown in Fig 1H–1J, the model successfully reproduced both the serial position and symbolic distance effects, confirming that our paradigm captures relational inference phenomena consistent with prior work (see S4B and S4C Fig, for Beta-Q and Betasort model fits). Together, these results indicate that participants infer relational knowledge from few-shot learning of local pairs, which could be well captured by standard computational models at the group level.

## Participants exhibit consistent, subject-specific local ranking errors

We next tested the first preregistered hypotheses by examining the consistent error patterns in participants' rankings after few-shot learning. Fig 2A shows their grand average ranking accuracy for all 28 local pairs. As expected, performance was higher for pairs with larger ranking distances (i.e., farther from the diagonal) and lower for those with smaller differences (closer to the diagonal).

Interestingly, although the group-level ranking behavior aligns with classical transitive inference and could be characterized by relevant computational models (see Fig 1), the pair-level accuracy across subjects reveals a different profile. Consider the 3rd versus 4th pair (Fig 2A, red box), with group mean accuracy of 0.67. The Independent-value account predicts a unimodal distribution centered near this mean, reflecting trial-by-trial fluctuation in noisy value estimates (Fig 2B, upper). The constructive ranking account predicts a bimodal distribution, with participants polarized into those consistently inferring 3rd < 4th versus 3rd > 4th (Fig 2B, lower).

Specifically, the accuracy showed a bimodal distribution across subjects (Fig 2B, lower), indicating that some participants consistently made correct inference (3rd < 4th), while others made robust errors (3rd > 4th), in line with the constructive ranking account. To quantify this bimodal pattern at pair-level, we fit a Beta distribution to the accuracy profile of each pair (Fig 2C). As shown in Fig 2D, easy pairs (large ranking distances; lower-left corner) followed high-accuracy profiles (brown, 13 out of 28 pairs; $\alpha > 1$, $\beta < 1$). Critically, difficult pairs (i.e., small ranking distance, near the diagonal) tended to show bimodal distributions (green, 15 out of 28 pairs; $\alpha < 1$, $\beta < 1$) instead of unimodal distribution (0 out of 28 pairs; $\alpha > 1$, $\beta > 1$). Notably, eight of the 77 subjects performed well (accuracy > 50%) on all local pairwise tests and were thereby excluded, leaving 69 subjects for pair-level analyses.

In addition to assessing cross-subject distribution at pair level, we further quantified the error consistency at the individual level. As shown in Fig 2E (left), participants exhibited high levels of error consistency (above 0.8; one-sample $t$ test, $p < 0.001$). Remarkably, 54 of the 69 participants (54/69, 78%) made 100% consistent errors on at least one local pair, and the number rose to 63 participants (63/69, 91%) for 80% error threshold (Fig 2E, right). The pre-registered results hold for both cohorts (see S1 and S2 Figs)

Notably, Q-learning simulations with matched accuracy failed to reproduce the observed bimodal patterns. Simulated difficult pairs yielded unimodal distributions (gray, Fig 2F versus green, Fig 2D), and simulated participants showed low error consistency with only 7 of 69 exceeding the 80% threshold (Fig 2G). Notably, despite its inability to account for idiosyncratic ranking structures, the Q-learning model is able to capture inter-individual differences in averaged accuracy, even under identical learning conditions. Beta-Q and Betasort models showed similar results (S4 Fig). Therefore, the computational models based on independent-value account, though successful at capturing group-level performance (see Fig 1H–1J), cannot account for the polarized, consistent error structure in individuals.

In summary, the findings support our *preregistered Hypothesis I* —bimodal distribution of local pair accuracy across subjects—indicating that despite receiving identical few-shot local training, participants exhibited stable, subject-specific errors on certain local pairs (see Fig 2H).

## Participants construct self-consistent, idiosyncratic global rankings

We next tested the *preregistered hypothesis II*, by conducting circular triads analysis to examine the self-consistent of local pairs with each individual. Specifically, as illustrated in Fig 3A, the relationships among three items (ground truth: A < B < C) can fall into one of three categories: correct triad (e.g., A < B, B < C, A < C; in line with A < B < C); self-consistent but incorrect triad (e.g., A < B, B > C, A < C; in line with A < C < B), and self-inconsistent and incorrect triad (e.g., A < B, B < C, A > C; violating transitivity). We applied this triplet-based analysis across all possible item triads for each participant to assess the internal consistency of their inferred rankings.

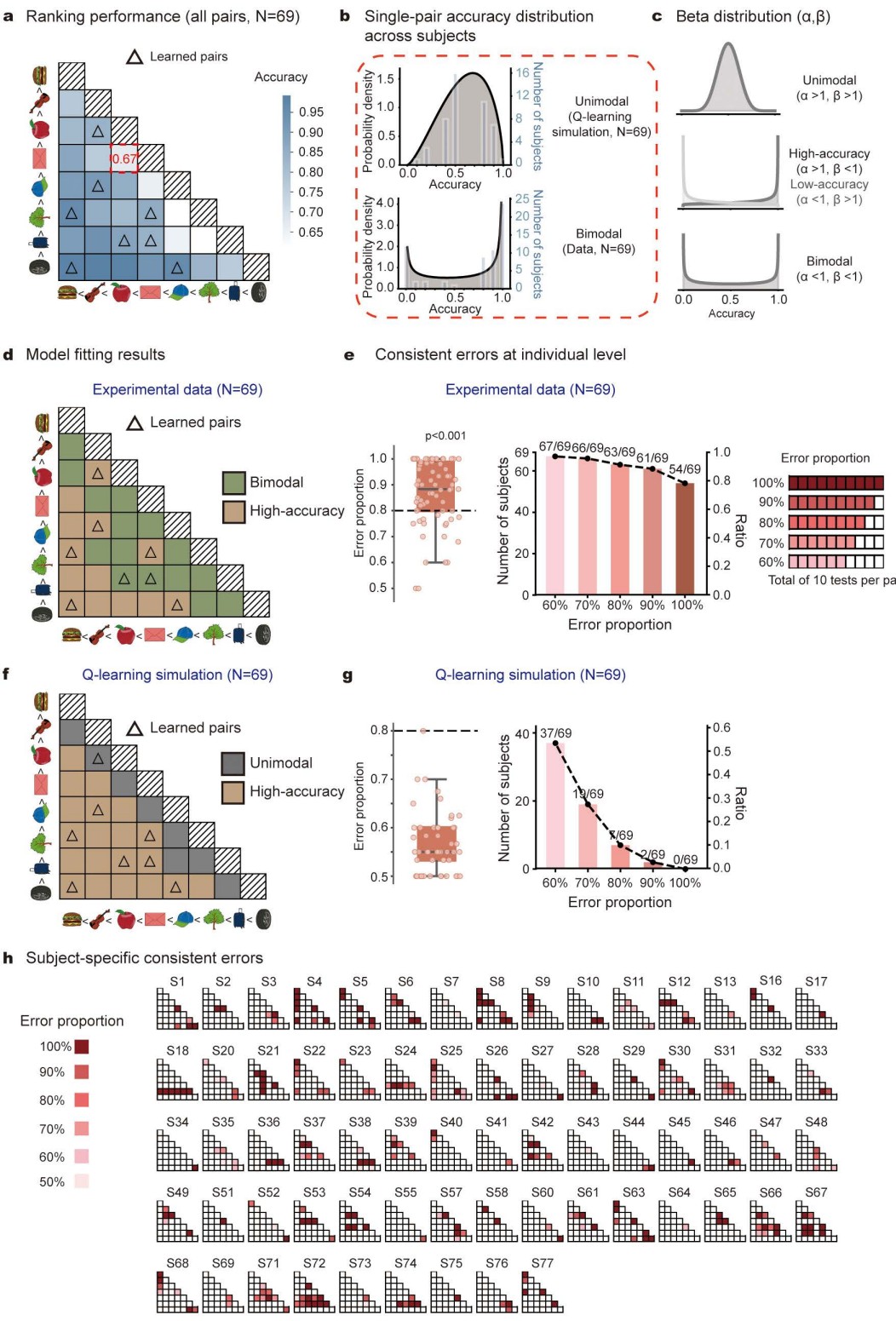

**Fig 2. Consistent, subject-specific local errors in testing phase (*preregistered hypothesis* I). (a)** Grand averaged accuracy matrix for all the 28 tested pairs, with row and column denoting corresponding items per pair (deep-to-light color represents high-to-low accuracy). Eight directly learned pairs (few-shot learning) were marked by triangles. Red dotted box highlights the exemplar pair in Fig 2B. **(b)** Illustration of unimodal and bimodal

accuracy distribution cross subjects for the exemplar pair (3rd vs. 4th) with a group mean accuracy of 0.67. *Upper*: unimodal distribution based on Q-learning model simulations (matched individual ranking accuracies), predicting converging global ranking across subjects. *Lower*: observed bimodal distribution, i.e., some subjects consistently made correct inference (3rd < 4th), while others made robust errors (3rd > 4th). **(c)** Beta-distribution model fitting of cross-subject accuracy distribution with two parameters ($\alpha$, $\beta$): Unimodal distribution (upper); High- or low-accuracy profile (middle); Bimodal distribution (bottom). **(d)** Beta-distribution model fitting results for each pair (brown: high-accuracy, $\alpha > 1$, $\beta < 1$; green; bimodal distribution, $\alpha < 1$, $\beta < 1$), consistent with *preregistered Hypothesis* I. **(e)** Individual-level error consistency. *Left*: grand averaged error consistency(>0.8, one-sample *t* test, $t = 4.406$, $p < 0.001$). dots denote individual data. *Right*: number and proportion of subjects making consistent error on at least one local pair for different thresholds (0.6 to 1.0). **(f)** Same as d but for Q-learning model simulation (brown: high-accuracy, $\alpha > 1$, $\beta < 1$; gray; unimodal distribution, $\alpha > 1$, $\beta > 1$). Note that simulation results support unimodal distribution that diverges from experimental findings. **(g)** Same as **e** but for Q-learning model simulation. Note the substantially decreased error consistency at individual levels. **(h)** Local error pattern for each subject. Blank tiles in the lower triangular matrices denote correct pairs (accuracy > 50%). Red tiles denote error pairs (deep-to-light color represents high-to-low error proportion). The data underlying this Figure can be found in https://osf.io/gya95/.

The Independent-value account predicts that the local errors reflect isolated, local distortions that occur randomly within each subject, leading to potentially large number of self-inconsistent local errors in individuals. In contrast, the constructive ranking account proposes that those local errors arise from a distorted yet coherent global ranking, therefore predicting the dominance of self-consistent local errors.

As previously noted, eight participants were classified into the correct-ranking group. Of the remaining 69, 64 (92.8%) exhibited self-consistent but incorrect rankings, whereas only 5 (7.2%) showed self-inconsistent errors (Fig 3B, left). Also, as preregistered, we quantified how self-consistent participants' judgements were by calculating the self-consistency coefficients based on how many self-inconsistent triads participants made (see Materials and methods). We tested the group-level self-consistency by shuffling the accuracy matrix of each participant. The participants showed significantly higher self-consistency than these from permutations ($p = 0.002$, permutation-based test; *the surrogation distribution, M* = 0.68, 95% quantile: 0.71; observed self-consistency coefficient: 1.00; from 0 to 1, from completely inconsistent to perfectly consistent). These results align with *preregistered Hypothesis II*, supporting that individuals made self-consistent local ranking errors. In other words, these local pairwise errors are not independent but are instead congruent within a global framework.

Furthermore, computational models based on the independent-value account failed to reproduce the observed self-consistency. We evaluated five models: three learning-based models that update item values from observed pairs (Q-learning with point estimates, BetaQ with distributional values, and Betasort with cross-item inference), a baseline model that averages presented distances (distance-averaging; S5 Fig), and a model using ground-truth values (correct-ranking; S5 Fig). All the models predominantly produced correct global rankings accompanied by self-inconsistent local errors, rather than the incorrect global rankings yet self-consistent local errors observed in the majority of human participants (Fig 3B, right and S6 Fig; see S11 Fig for deterministic simulation of Q-learning).

Finally, we reconstructed each participant's subjective ranking based on their own 28 pairwise judgments using the HodgeRank method (see Materials and methods; example subject shown in Fig 3C). This analysis revealed individualized global rankings that deviated from the ground-truth ordering, resulting in low inter-individual similarity (Fig 3E). In contrast, simulations of independent-value computational models produced substantially higher inter-subject similarity (Fig 3D; Q-learning, two-sample *t* test, $t = -66.77$, $p < 0.001$; see S6 Fig for results of other models), indicating that the classical models predict convergence toward a shared ranking rather than the observed idiosyncratic rankings.

In summary, these findings support the constructive ranking account and confirm preregistered Hypothesis II: participants construct self-consistent, idiosyncratic global rankings that deviate from the ground-truth order.

## MEG experiment and replication of behavioral findings

Finally, we conducted MEG recordings using an Elekta NEUROMAG system in a new cohort of participants ($N = 30$; three excluded due to below-chance learning performance) to examine the neural basis of subject-specific global rankings formed through few-shot learning.

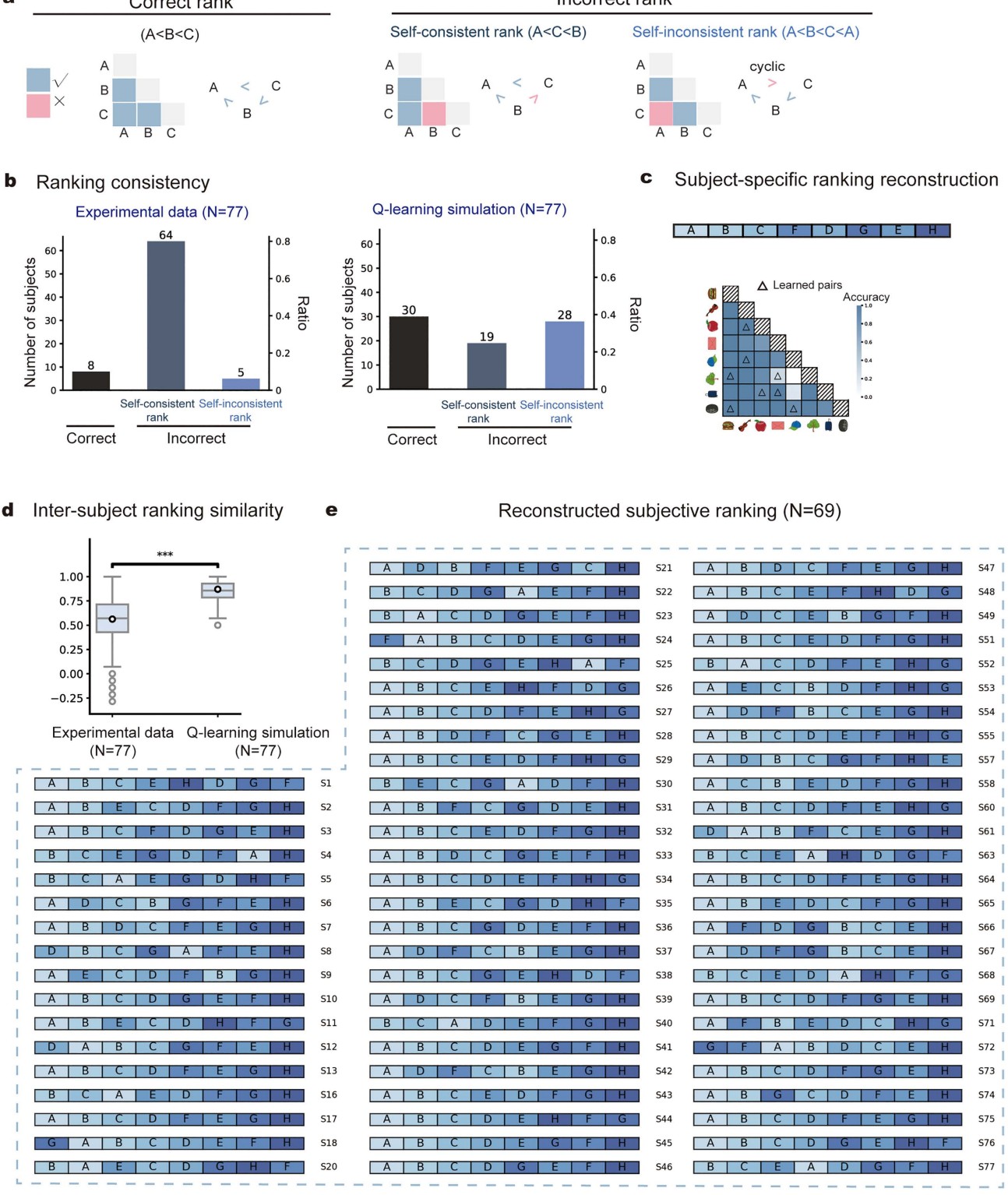

**Fig 3. Self-consistent, subject-specific global rankings (*preregistered hypothesis* II). (a)** Illustration of self-consistency ranking analysis for a triple sequence (A<B<C). Left: correct ranking (A<B, B<C, A<C; consistent with ground truth); Middle: self-consistent but incorrect ranking (A<B, B>C, A<C; consistent with A<C<B); Right: self-inconsistent and incorrect ranking (A<B, B<C, A>C; violating transitivity). The triplet-based analysis was

conducted across all possible triads to assess the internal consistency of inferred rankings. **(b)** *Left:* Number of subjects for each self-consistency category. *Right:* Number of subjects for each self-consistency category based on Q-learning model simulation. **(c)** Exemplar subjective ranking reconstruction. *Upper:* reconstructed individualized global ranking. *Lower:* corresponding accuracy matrix for this subject. Triangles mark the directly learned pairs. **(d)** Inter-subject ranking similarity for experimental data and Q-learning model simulation (two-sample $t$ test, $p < 0.001$). Q-learning simulations showed higher inter-subject similarity in global rankings than behavioral result. **(e)** Reconstructed subjective-specific rankings for all the 69 subjects (eight correct-ranking participants excluded). The data underlying this Figure can be found in https://osf.io/gya95/.

The MEG experiment comprised three phases. In the pre-learning phase, participants performed a sequential comparison task on eight images, judging whether each item was larger or smaller in real-world size than the preceding item (5 blocks × 81 images, yielding 80 pairwise judgments per block; no feedback). Next, participants completed the few-shot learning phase, identical to that used in the preregistered behavioral experiment. In the post-learning phase, participants performed the same sequential comparison task as in the pre-learning phase, but based on the newly learned ranking. Thus, the pre- and post-learning phases employed identical stimuli, timing, and task demands, differing only in the underlying ranking rule. This design offers two advantages. First, the pre-learning phase served as a within-subject baseline, capturing stimulus-driven neural responses and pre-existing representational structure. Second, contrasting post-learning with pre-learning activity isolated neural reorganization specifically attributable to newly constructed rankings, while controlling for stimulus properties and individual differences in baseline activity.

All key behavioral findings were replicated. Participants performed well on both ranking tasks (Fig 4C) and exhibited the classic serial position and symbolic distance effects (S7 Fig). Importantly, 24 of the 27 participants (88%) made highly consistent errors (≥80%) on at least one local pair (Fig 4D), and 26 participants exhibited self-consistent but incorrect global rankings (Fig 4E). Behavioral patterns in the MEG cohort closely matched those observed in the preregistered behavioral experiments (S7 Fig). Furthermore, the constructed rankings were independent of real-world size knowledge (mean $r = 0.03$, 95% CI [−0.06, 0.13]; S10 Fig), indicating that the pre-existing semantic structure did not impact learned rankings.

### Neural activities align with idiosyncratic global rankings after few-shot learning

We hypothesized that neural correlates of subject-specific global rankings would emerge following few-shot learning in the post-learning phase, whereas real-world size representations—being innate—would be present in both pre- and post-learning phases. To characterize the neural basis of constructed rankings, we employed representational similarity analysis (RSA), which assesses whether neural response patterns reflect hypothesized relational structures by testing whether items that are closer along a given dimension evoke more similar neural activity.

Three model representational dissimilarity matrices (RDMs) were constructed (see Fig 5A), capturing low-level visual features, real-world size rankings, and subjective rankings derived from individual behavioral data, respectively. At each time point, we performed multiple linear regression analyses in which each participant's neural RDM served as the dependent variable and the three model RDMs were entered simultaneously as predictors (Fig 5A). This procedure yielded participant-specific $\beta$ coefficients that quantified the strength of neural encoding for each relational structure. For real-world size, we averaged the corresponding $\beta$ coefficients across the pre- and post-learning phases, reflecting its persistent representation as innate knowledge. Critically, for subjective ranking, we computed the difference between post- and pre-learning coefficients, $\beta(\text{post}) - \beta(\text{pre})$, to isolate neural reorganization specifically attributable to few-shot learning. This controls for pre-existing alignment between intrinsic neural geometry and the to-be-learned structure, ensuring that the observed effects reflect learning-induced changes rather than baseline correlations.

Results revealed temporally and spatially dissociated neural signatures. Pixel-level visual similarity emerged early (165–290 ms) in both pre- and post-learning phases (Fig 5B; cluster-based permutation test, $p = 0.001$), confirming stimulus-driven processing independent of tasks. Real-world size representations appeared at intermediate latencies

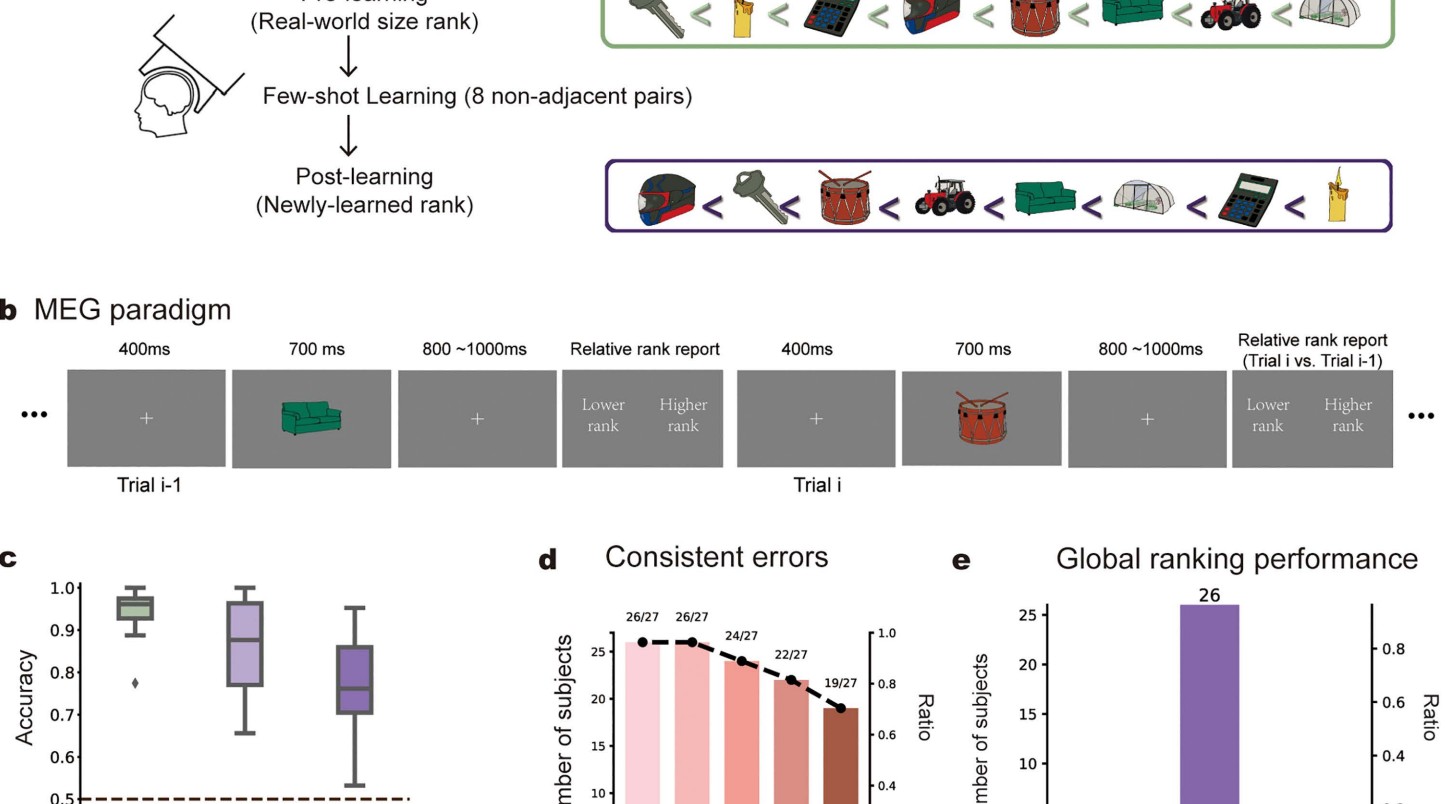

**Fig 4. Experimental design and behavioral results (MEG experiment). (a)** MEG experiment consists of three phases—pre-learning, few-shot learning, and post-learning. In the pre-learning phase, participants performed a real-world size ranking task on eight images (e.g., Sofa is larger than Drum). Next, they learned new ranks for the same images through eight non-adjacent local pairs (Few-shot learning), and tested in the post-learning phase (e.g., Sofa ranks higher than Drum). **(b)** MEG experimental procedure. Participants judged on each trial whether the current item was larger or smaller than the previous one in terms of real-world size (pre-learning phase) or newly learned ranking (post-learning phase). Note that pre- and post-learning phases shared the same stimuli and procedure, only differing in ranking rule. **(c)** Boxplot of grand averaged ranking accuracy for real-word size task (pre-learning phase; green) and newly learned ranking test (post-learning phase; light purple: learned pairs; dark purple: non-learned pairs). **(d)** Number and proportion of subjects making consistent error on at least one local pair for different thresholds. **(e)** Number of subjects for each self-consistency category. The data underlying this Figure can be found in https://osf.io/gya95/.

(240–480 ms) with comparable strength across both phases (Fig 5C; cluster-based permutation test, 240–480 ms, $p = 0.001$; 710–1,015 ms, $p = 0.001$). Crucially, subjective ranking representations emerged substantially later (685–775 ms, $p = 0.041$) and occurred only during post-learning (Fig 5D). A permutation test, shuffling subjective ranking RDM across subjects, confirmed the subject-specific nature of global rankings—neural responses aligned with each subject's own ranking, not with others' (Fig 5F; $p = 0.01$). Sensor-level searchlight analysis revealed distinct spatial distributions: low-level features in right parieto-occipital cortex, real-world size at posterior and lateral sensors, and subjective ranking localized to mid-parietal regions (Fig 5E).

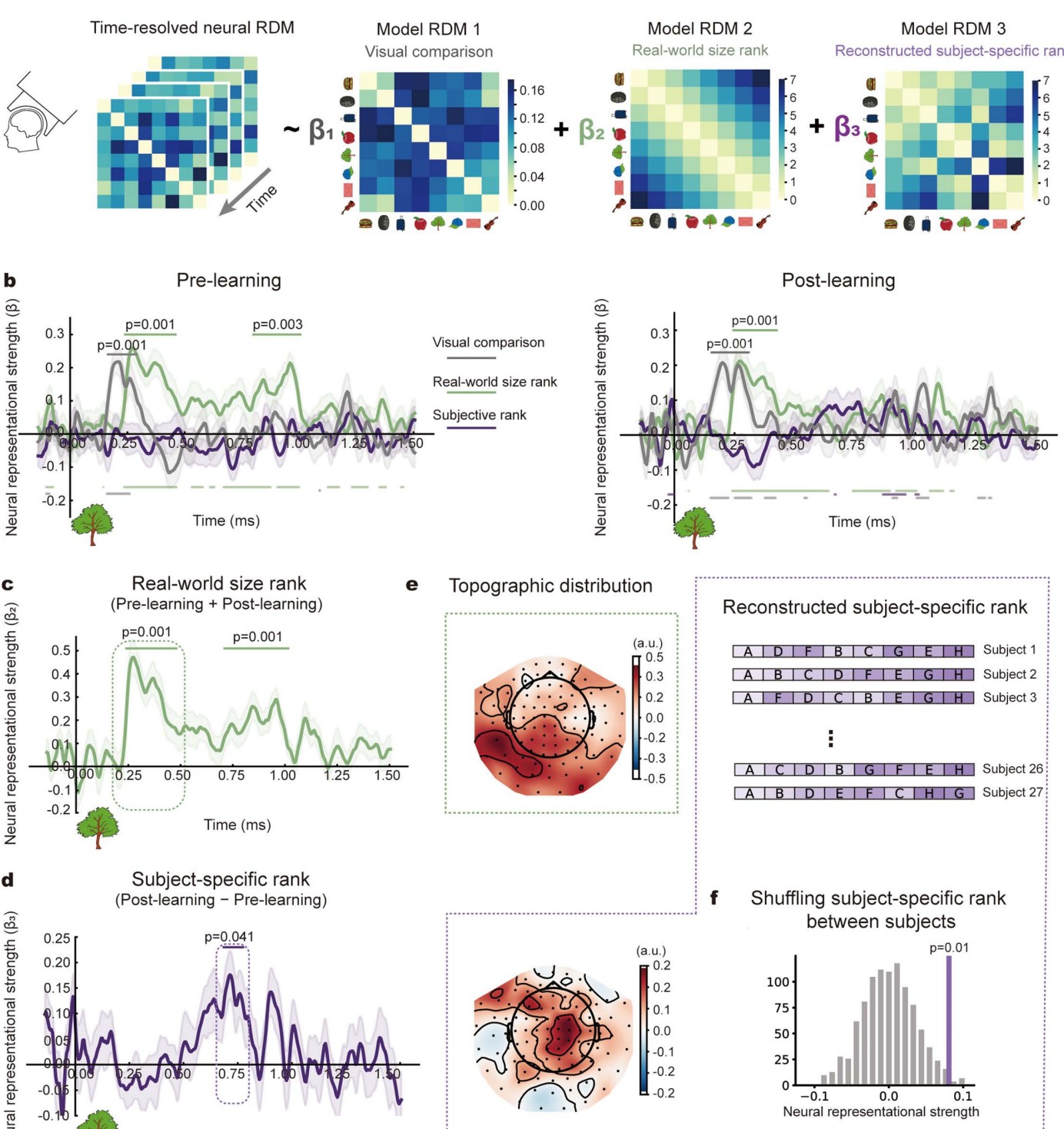

**Fig 5. Neural correlates of subjective ranking after few-shot learning (MEG experiment). (a)** Three representational dissimilarity matrices (RDMs) were constructed, based on low-level visual similarity (RDM1; gray; identical across subjects), real-world size (RDM2; green; identical across subjects), and each subject's reconstructed global ranking (RDM3; purple), respectively, which were used to regress neural dissimilarities, separately for pre- and

post-learning phases. **(b)** Grand-averaged time-resolved RSA results during pre-learning (left) and post-learning (right) for three RDMs (gray: low-level features; green: real-world size; purple: subjective ranking). The horizontal lines indicate corresponding significant time ranges (upper, cluster-based permutation test, $p < 0.05$; lower, one-sample $t$ test, $p < 0.05$ uncorrected). Shaded area denotes cross-subject standard error. **(c)** Grand averaged time-resolved RSA results for real-world size as a function of time after image onset. Horizontal lines denote significant time window (cluster-based permutation test, $p < 0.05$). **(d)** Same as **c** but for newly-learned ranking. **(e)** Topographic distributions for real-world size rank (upper) and subjective rank (bottom) using sensor-level search light analysis. **(f)** Permutation test ($n = 1,000$, $p < 0.05$) examining neural correlates of subject-specific global ranking (dotted purple box in **d**) by shuffling RDM2 across subjects (upper). The data underlying this Figure can be found in https://osf.io/gya95/.

Notably, the subjective ranking effect during the post-learning phase, when analyzed without pre-learning baseline correction, shows a nominally significant peak within a similar temporal window (Fig 5B; peak at 890 ms, $t(26) = 2.80$, uncorrected $p = 0.005$), but does not survive cluster-based permutation correction ($p = 0.13$), indicating reduced sensitivity when geometric confounds are not controlled. Moreover, the subjective ranking effect does not reach significance when the three excluded participants with below-chance behavioral performance are included ($p = 0.089$, cluster-based permutation test). This pattern is consistent with the expectation that unsuccessful learners introduce noise into the data, thereby reducing sensitivity to learning-related effects.

Together, these results provide converging neural evidence that subjects construct idiosyncratic, coherent global rankings after few-shot learning of local pairs.

## Discussion

Here we investigated how humans perform relational inference under few-shot local learning constraints. We hypothesized that relational inference is shaped by inductive biases, leading individuals to construct stable, self-consistent, yet idiosyncratic rankings that deviate from the ground-truth order, even when exposed to identical local inputs. These hypotheses were preregistered and confirmed in both behavioral experiments and an independent MEG study. MEG recordings further showed that neural similarity patterns between items are reorganized to align with each individual's subjective ranking, rather than those by others. This subject-specific representation emerges in frontoparietal cortex and is distinct from neural signatures associated with innate ranking knowledge. Together, our findings support the active construction account, underscoring the constructive nature of human relational learning—when faced with sparse samples and limited computational resources, the human brain actively infers and imposes structure.

Our findings largely replicate classical phenomena in transitive inference at the group level, including the canonical serial position and symbolic distance effects—hallmark signatures of relational learning observed across species and paradigms [20,22]. Notably, these effects emerged despite a fundamental departure from traditional designs: whereas classical paradigms rely on extensive training with adjacent pairs and continuous feedback [16,17], our participants received only brief few-shot exposure to largely non-adjacent pairs, followed by testing without feedback. Importantly, these group-level effects could also be well captured by classical computational models based on incremental value updating mechanisms [9,27,34,35].

Beyond replicating canonical effects, our results substantially extend previous work on transitive inference by revealing stable, self-consistent, yet idiosyncratic ranking patterns at the individual level. These individual-level structures cannot be readily explained by classical value-updating models, highlighting a dissociation between group-level regularities and individual-level inference. This divergence suggests that incremental value updating alone is insufficient to capture the full nature of relational inference, particularly under few-shot conditions. Rather than passively accumulating evidence and updating items independently, the observed global self-consistency across local pairs points to the active imposition of a relational schema during learning. This *constructive ranking account* aligns with theoretical frameworks emphasizing structural priors [11,36] and geometry-based representations of ranking [1,30,37,38]. From this perspective, relational learning does not merely tune local values, but involves the construction of a global relational structure that constrains and supports internal consistency within each individual.

Numerous models have been proposed to explain transitive inference. One class posits that inference relies on an internal "number line" representation of items, enabling implicit global updates on each trial [37], and another models transitive learning as reinforcement learning with asymmetric update rules [16]. Moreover, transitive inference has been quantified as probabilistic preference learning, e.g., via the one-parameter Mallows model, which infers rankings from the distribution of possible global orderings given observed pair [30]. While our findings are not readily explained by these models, they align with the broader notion that even isolated pairwise learning triggers the construction of a unified mental number line that organizes and updates items globally, yielding internally consistent yet incorrect subjective rankings. In this view, value updating is not confined to local pairs but propagates automatically across the entire set, constrained by the inferred relational structure.

One might argue that idiosyncratic rankings simply reflect attentional fluctuations, imperfect encoding, or random forgetting across individuals—factors that naturally arise under limited computational resources, particularly in few-shot learning. However, the observed pattern of self-consistent errors cannot be explained by merely introducing stochasticity or idiosyncratic attentional fluctuations into traditional independent-value frameworks. Critically, the remarkably low rate of circular triad errors in local pairwise comparisons indicates that errors are not independent noisy distortions, but are instead systematically constrained by each individual's subjectively constructed global ranking. This global structure cannot be captured by stochastic attentional parameters acting independently on each local pair. Our results therefore point to a mechanism largely absent from current frameworks—the brain actively imposes a global relational structure to maintain internal coherence, even at the cost of making systematic errors on specific local comparisons. Instead of passively accumulating independent pairwise evidence, participants construct unified ranking schemas guided by inductive biases—internal constraints that shape the hypothesis space and enforce transitivity. This constructive process underscores the active, generative nature of human relational learning under sparse evidence and limited cognitive resources.

Our MEG data show that item-wise neural similarity patterns are reconfigured to reflect each participant's idiosyncratic ranking. This newly acquired ranking representation emerges later in the frontoparietal cortex, whereas innate knowledge rankings (e.g., real-world object size) appear earlier in temporal regions. Prior neurophysiological studies in monkeys and neuroimaging work in humans have established the central role of the frontoparietal cortex in transitive inference [2,39–44]. More recently, a "list linking" fMRI study reported rapid reorganization of object representations along an elongated neural manifold in dorsal frontoparietal areas following linking-pair exposure [1]. It has also been proposed that neural representations along a "geometric" mental line arise as a monotonic mapping between sensory input and behavioral output, suggesting a possible role for motor planning in transitive inference [37]. Our findings are consistent with these accounts, reinforcing the pivotal role of the frontoparietal cortex in storing newly learned relationships. Our results further extend these accounts by demonstrating that neural codes reflect the outcome of an idiosyncratic constructive process, rather than shared or canonical knowledge.

A recent study reported that monkeys exhibit superstitious learning by imposing spurious orderings on untrained stimulus sets, irrespective of random reward feedback [45]. This subjective ranking has been interpreted as a model-based strategy operating in unstructured environments, with internal models formed independently of trial-and-error learning. Our findings parallel but also diverge from this account. Unlike the random, unstructured relationships examined previously, our stimuli contained a hidden ranking structure that participants had to infer from limited exposure to partial pairs. Moreover, no feedback was provided, requiring pure inference rather than reinforcement. These results extend prior work by showing that subjective ranking is not restricted to unstructured inputs but also emerges in structured contexts, where it can induce idiosyncratic distortions of the true ranking.

What determines the particular "incorrect" global ranking adopted by each subject? Several factors may contribute when individuals attempt to learn relational information beyond their processing capacity. Some research indicates that learning sequence may influence the outcome [46], yet both correlational analyses and a controlled between-group experiment found no evidence that learning sequence order drives the observed inter-individual variability in subjective rankings

(S8 and S9 Figs). Participants may also differentially weight item features, using them as idiosyncratic anchors to impose latent structural templates during relational inference [39,47]. They may also rely on distinct memory compression strategies, storing a reduced and simplified version of the relational graph, which inevitably introduces systematic distortions or biases [1,30]. Consistent with this view, a recent neural network model of transitive inference showed that variation in a single conjunctive parameter in the output layer could produce individual differences in ranking behavior [6]. Additionally, attentional fluctuations and encoding noise—natural consequences of learning under limited computational resources— may further shape which particular ranking structure each individual settles on, by determining which pairs are most reliably encoded and which are left to constructive inference. In this sense, limited computational resources and encoding noise may set the conditions under which constructive ranking is engaged, while the constructive mechanism itself is what ultimately enforces internal coherence across local pairs.

In conclusion, our study highlights the constructive and generative nature of relational learning when faced with sparse inputs, whereby incomplete and noisy inputs are re-registered into an internally constructed, individually idiosyncratic ordering schema.

## Materials and methods

### Participants

In the behavioral experiment, we initially recruited 40 participants through online social media platforms of Peking University as specified in our preregistration, confirming our preregistered hypothesis. To further validate and replicate our initial findings, we recruited an additional 40 participants (total $N = 80$). Three participants were excluded for performing below chance level (50% accuracy) across all 10 testing blocks, resulting in a final sample of 77 participants. For the MEG experiment, we recruited 30 participants from the same population. Three participants were excluded for below-chance performance, yielding a final sample of 27 participants.

The study was approved by the Institutional Review Board of School of Psychological and Cognitive Sciences at Peking University (#2021-10-15), according to the principles expressed in the Declaration of Helsinki. Participants provided written informed consent and received monetary compensation for their time (45 CNY for behavioral experiments, 250 CNY for MEG experiment).

### Behavior experimental design

In the experiment, participants were asked to learn a ranking sequence of 8 items (A<B<C<D<E<F<G<H) in the learning phase and were tested in the subsequent test phase. Item images were manually selected from online open-access resources [48]. The same eight items were reused across participants while the ranking sequence of each participant was randomly shuffled. Fig 1B illustrates an example sequence for one participant.

In each trial of the learning phase, two items with two ticked bars were presented to participants. Participants were instructed that each item represented a movie and that they needed to rank the movie by popularity according to their box office performance. The ticked bars showed how much the two movies sold in one single day. Participants were explicitly instructed to focus on the relative difference of the bars instead of the absolute values as box office selling is subject to day-by-day fluctuation. Participants need to rank the popularity of the movies according to the item pairs. Participants did not need to respond in the learning phase. Eight non-adjacent item pairs were presented for participants.

In the learning phase, participants learned a fixed set of eight non-adjacent item pairs, identical across all participants: (A,F), (B,C), (B,E), (C,G), (D,F), (D,G), (E,H), (A,H). Each item appeared in two pairs equally. The relative differences of a pair ranged from 1 to 7 units (Fig 1B and 1C). Although relative differences were determined by item relationships in the rank, the absolute values of bars were determined randomly each trial. The learning phase comprised 4 continuous blocks, with each of the 8 pairs presented exactly once per block in a randomized order. This design ensured that

participants received identical few-shot exposure (4 presentations per pair) while controlling for sequence effects through randomization within each block.

Following the learning phase, the testing phase consisted of 10 continuous testing blocks. In each block, a participant completed all $\binom{8}{2}$ = 28 possible pairwise comparisons in a random sequence. In each trial, a participant made a comparison by clicking on the item with the higher rank while two items were presented in parallel on the screen. The next trial starts after 2-s intertrial interval. Critically, no feedback was provided during the testing phase.

Behavioral experiments were implemented with PsychoPy and hosted on Naodao and Credamo. Participants were provided with the experiment link and completed the experiment on their own personal computer/laptop.

## Behavioral analysis

**Beta distribution fitting.** To distinguish whether participants' accuracy patterns reflected polarized responses (clustering near perfect accuracy or complete errors) versus normally distributed performance around an intermediate value, we employed Beta distribution fitting. We fitted a Beta distribution parameterized by shape parameters ($\alpha$, $\beta$) to the accuracy distribution for each pair across all participants. Based on the values of $\alpha$ and $\beta$, we classified accuracy distributions into four categories: unimodal, high-accuracy, low-accuracy, and bimodal. When both $\alpha > 1$ and $\beta > 1$, the distribution exhibited a unimodal bell-curve shape centered around an intermediate accuracy level. When $\alpha > 1$ and $\beta < 1$, the distribution showed a high-accuracy pattern with right-skewed distributions toward high performance, while $\alpha < 1$ and $\beta > 1$ produced low-accuracy patterns with left-skewed distributions toward poor performance. When both parameters fell below 1 ($\alpha < 1$, $\beta < 1$), the distribution became bimodal with probability mass concentrated at the extremes (0 and 1), indicating that participants predominantly exhibited either consistently correct or consistently incorrect responses for those pairs. For our analysis, we focused primarily on distinguishing between unimodal and bimodal patterns, as the high-accuracy and low-accuracy conditions could not reliably indicate whether participants exhibited bimodal or unimodal error patterns due to ceiling and floor effects.

**Consistent errors.** We quantified the error consistency of a participant by calculating the proportion of consistent errors among all pairs for the participant. A pair is defined as a consistent error pair if the proportion of error trials exceeds a threshold. As we have 10 trials for each pair, the threshold has 5 levels: 60%, 70%, 80%, 90%, and 100%. We first use the strictest threshold 100%, i.e., an error pair is treated as consistent only when a participant made wrong choices in all 10 trials for the pair. We calculated the proportion of pairs with consistent errors for each participant. The proportions under different thresholds were also calculated (Fig 2E). For the reference level, we use a threshold of 80%, which still represented high consistency, a conservative threshold that did not compromise our findings.

**Ranking self-consistency.** After categorizing the pair-wise judgement, we can symbolically examine whether the ranking judgements of a participant are self-consistent. Any item triad can fall into one of the following three categories (Fig 3A) by the consistency of the three judgements: correct triad (e.g., A<B, B<C, A<C; in line with A<B<C); self-consistent but incorrect triad (e.g., A<B, B>C, A<C; in line with A<C<B), and self-inconsistent and incorrect triad (e.g., A<B, B<C, A>C; violating transitivity). We quantified ranking consistency through the self-consistency coefficient, defined as:

$$1 - \frac{N_d}{N_T}$$

(1)

where $N_d$ represents the number of self-inconsistent and incorrect triads; $N_T = 20$ denotes the maximum possible circular triads for 8 items. The maximum number of the circular triads in a set of $n$ elements is defined by two distinct formulas, depending on whether n is odd or even: $(n^3 - n)/24$ and $(n^3 - 4n)/24$, respectively [49]. The self-consistency coefficient ranges from 0 (complete inconsistency) to 1 (perfect consistency).

Permutation test for self-consistency coefficient is comparing the group-level self-consistency coefficient with a surrogation distribution. We broke the self-consistency by shuffling the pairwise accuracy matrix of each participant. The self-consistency coefficient was then calculated based on the shuffled accuracy matrix. After 1,000 permutations, we compared the group-level self-consistency with the surrogation distribution.

The ranking of a participant was classified as globally self-consistent ranking if and only if its self-consistency coefficient equaled 1 (i.e., $N_d = 0$). This strict criterion follows from transitive logic: any single circular triad (Fig 3A, right) suffices to establish global inconsistency, as larger cycles (e.g., A<B<C<D<A) necessarily contain at least one such circular triad (e.g., A<B<C<A).

**Ranking reconstruction.** For participants with self-consistent rankings, a single deterministic global ranking can be directly derived from their pairwise judgments. For those participants, we inferred their subjective rankings using the Hodge-rank method [50]. The Hodge-rank method aims to find optimal global ranking scores $s$ for each item by minimizing the following objective function:

$$\min_{s \in R^8} \sum_{i,j} \left( s_i - s_j - \hat{Y}_{ij} \right)^2$$

(2)

where $\hat{Y}_{ij}$ represents the average preference score across all 10 trials for each pair $(i, j)$. For each trial, $\hat{Y}_{ij} = 1$ if the participant chose item $i$ over item $j$, and $\hat{Y}_{ij} = -1$ if the participant chose item $j$ over item $i$. The individual ranking for each participant was determined by sorting the items according to their computed ranking scores $s$.

**Inter-subject ranking similarity.** To quantify inter-subject ranking similarity, we computed Kendall's tau correlation coefficient between the global ranks of every two participants of all 69 participants. To compare ranking consistency between empirical data and computational modeling, we statistically contrasted two distributions: (1) the inter-subject similarity distribution derived from participants' reconstructed global ranks, and (2) the corresponding distribution generated by model simulations, using independent samples $t$-tests.

## Preregistration and deviations

We pre-registered the behavioral experiment on the Open Science Framework prior to data collection (https://osf.io/dpxq4). In this pre-registration, we specified our novel behavioral paradigm, target sample size ($N = 40$), measured variables, and statistical analysis. Specifically, we preregistered two core hypotheses: Hypothesis I predicted bimodal accuracy distributions across subjects and pairs, tested via Beta distribution fitting to individual participants' accuracy profiles ($\alpha < 1$ and $\beta < 1$). Hypothesis II predicted globally self-consistent rankings within individuals, tested via permutation-based self-consistency analysis. Participants of below-chance average accuracy (<0.5) were excluded from analysis as pre-registered.

After confirming both hypotheses in the pre-registration cohort ($N = 40$), we recruited an independent replication cohort ($N = 40$; three participants excluded due to below-chance learning performance) to strengthen the robustness of our findings. Although we report combined results ($N = 77$) in the main text for narrative clarity, both cohorts independently confirmed both preregistered hypotheses when analyzed separately (S1 and S2 Figs).

**Deviation from preregistration.** In calculating the self-consistency coefficient (Equation 1), our preregistration used all possible triads ($N_T = 56$) as the denominator. However, subsequent theoretical analysis revealed that for 8 items, the maximum number of circular triads is 20, not 56 [49]. We therefore revised the denominator to $N_T = 20$ to accurately reflect the theoretical maximum.

**Exploratory analyses.** Beyond the pre-registered individual-level Beta distribution fitting (Hypothesis I), we conducted two additional analyses to provide complementary perspectives on error patterns without introducing alpha inflation concerns: (1) pair-level Beta distribution fitting across subjects (Fig 2D versus 2F), revealing bimodal distributions

for difficult pairs at the group level, and (2) error consistency analysis (Fig 2E and 2G), quantifying the proportion of consistently erroneous pairs within individuals.

The MEG experiment was a separate, non-preregistered, follow-up study conducted with an independent cohort after the behavioral experiments.

Prior to formal hypothesis pre-registration, we conducted an exploratory behavioral experiment ($N = 35$, Peking University participants) using the same learning phase procedure as would later be preregistered, but with a reduced testing phase assessing each pair only 5 times. This preliminary study informed our pre-registered hypothesis regarding consistent errors and guided sample size determination.

## Modeling

**Q-learning model.** In Q-learning model [16,51], we assume that participants update the ranking value of each item using a simple delta rule during learning phase. Specifically, values are updated upon presentation of a learned pair. During the learning phase, when a pair ($m$, $n$) is presented, their values ($Q$) are updated as follows:

$$Q_m \leftarrow Q_m + \alpha \left[ D(m, n) - (Q_m - Q_n) \right] /2, \tag{3}$$

$$Q_n \leftarrow Q_n + \alpha \left[ D(n, m) - (Q_n - Q_m) \right] /2, \tag{4}$$

where free parameter $\alpha$ denotes the learning rate ($0 < \alpha \leq 1$). The presented relative difference $D(m, n)$ between the $m$-th-ranked item and $n$-th-ranked item is quantified as:

$$D(m, n) = (m - n)/7, \tag{5}$$

The initial values of all $Q$s were set to 0.

During the testing phase, when the pair ($i$, $j$) is presented, the item $i$ is selected with a probability defined by a *softmax* function with an inverse temperature parameter $\gamma (\gamma \geq 0)$:

$$p_{i,j}(i) = \frac{1}{1 + \exp\left(-\gamma \left(Q_i - Q_j\right)\right)} \tag{6}$$

where $Q(\cdot)$ denotes the final $Q_t(\cdot)$ after the learning phase.

**Beta-Q model.** Beta-Q model involves beta distribution updating in learning [17]. The ranking value of each item was assumed to follow a beta distribution, $\mathrm{Beta}(U, L)$, with two shape parameters, $U$ and $L$ for each item. The probability density function of $\mathrm{Beta}(U, L)$:

$$\mathrm{Beta}(x; U, L) = \frac{\Gamma(U + L)}{\Gamma(U)\Gamma(L)} x^{U-1}(1 - x)^{L-1} \tag{7}$$

Before learning, the initial values of $U_i$ and $L_i$ were set to 1 as $\mathrm{Beta}(1, 1)$ is a uniform distribution between 0 and 1. When a pair ($m$, $n$) is presented at the learning phase ($m > n$), the ranking value distributions are updated by adjusting $U$ and $L$ as follows:

$$U_m \leftarrow U_m + \left[ D(m, n) - \Delta V(m, n) \right], \tag{8}$$

$$L_m \leftarrow L_m + \alpha \left[ D(n, m) - \Delta V(n, m) \right],\tag{9}$$

$$U_n \leftarrow U_n + \alpha \left[ D(n, m) - \Delta V(n, m) \right],\tag{10}$$

$$L_n \leftarrow L_n + \left[ D(m, n) - \Delta V(m, n) \right],\tag{11}$$

where $\alpha$ is a bias factor toward boundary ($0 < \alpha \leq 1$); $\Delta V(m, n)$ is the expected value difference between $m$ and $n$,

$$\Delta V(m, n) = V_m - V_n,\tag{12}$$

where $V = U/(U + L)$ is the expected value of an item.

During the testing phase, when the pair $(i, j)$ is presented, the judgment is made by independently sampling values from $\mathrm{Beta}\left(U_i, L_i\right)$ and $\mathrm{Beta}\left(U_j, L_j\right)$ for item $i$ and $j$, respectively, and then comparing the two drawn values. Thus, the probability of choosing item $i$ over $j$ is given by

$$p_{i,j}(i) = \int_0^1 \mathrm{Beta}\left(x \mid U_i, L_i\right) \int_0^x \mathrm{Beta}\left(y \mid U_j, L_j\right) \mathrm{d}y\, \mathrm{d}x.\tag{13}$$

**Beta-sort model.** The beta-sort model is an extension of beta-Q model [17]. The additional assumption beta-sort model is that the learning, or value updating, not only takes place at the items being presented but also at those which are not presented. When a pair $(m, n)$ is presented ($m > n$), the ranking value distribution of item $m$ and $n$ are updated as in beta-Q (Equations 5–8). For items that are not presented, their updating is as follows.

For ranking value distributions whose expected values fall between $V_m$ and $V_n$, their variances shrink while their means keep the same, as the current observation confirms the values. For any item $o$ which satisfies $V_n < V_o < V_m$,

$$U_o \leftarrow U_o + V_o,\tag{14}$$

$$L_o \leftarrow L_o + 1 - V_o.\tag{15}$$

For items with lower expected ranking value, their distributions are pushed downward. For any item $o$ which satisfies $V_o < V_n$,

$$U_o \leftarrow U_o + \Delta V(m, n),\tag{16}$$

$$L_o \leftarrow L_o + \alpha \Delta V(m, n).\tag{17}$$

Similarly, for items with higher expected ranking value, their distributions are pushed upward. For any item $o$ which satisfies $V_o > V_m$,

$$U_o \leftarrow U_o + \alpha \Delta V(m, n),\tag{18}$$

$$L_o \leftarrow L_o + \Delta V(m, n).\tag{19}$$

Choice policy is the same as beta-Q model.

**Distance-averaging model.** The distance-averaging model serves as a non-learning baseline that computes item values by averaging the signed distances across all learned pairs in which each item appears. Unlike learning models, this model does not perform trial-by-trial updates during the learning phase.

For each item $i$, the final value $Q_i$ is computed as:

$$Q_i = \frac{1}{n_i} \sum_{j \in P_i} D(i,j)$$

(20)

where $n_i$ denotes the number of learned pairs containing item $i$, $P_i$ represents the set of items paired with item $i$ in the learning phase, and $D(i,j)$ represents the signed distance between items $i$ and $j$ as defined in Equation 5. For example, if item $m$ appears in two learned pairs with signed distances of +3 and +7, its value would be $Q_m = ((+3) + (+7))/2 = +5$.

During the testing phase, choices are generated using the same softmax policy as the Q-learning model (Equation 6).

**Correct-ranking model.** The correct-ranking model serves as an oracle baseline that directly uses ground-truth ranking values without any learning process. Each item is assigned its true rank position:

$$Q_i = r_i$$

(21)

where $r_i \in \{1, 2, \ldots, 8\}$ represents the ground-truth rank of item $i$. During the testing phase, the model employs the same softmax choice policy (Equation 6).

**Model fitting and simulation.** We fitted the models to participants' trial-by-trial choices in the testing phase through maximum likelihood estimation for each participant. Optimization was implemented with *optimize.minimize*() function of *SciPy* package in Python. Using the best-fitting parameters and the learning sequence for each participant, we subsequently randomly generated participants' trial-by-trial choices. Behavioral analyses were conducted on these model-simulated responses the same as on participants data, enabling direct quantitative comparison between the model's predictions and the observed human behavior.

## MEG experimental design

The MEG experiment comprised three sequential phases conducted under continuous MEG recording: a pre-learning testing phase, a few-shot learning phase, and a post-learning testing phase. We used a different set of eight images, identical across all participants.

The learning phase replicated the behavioral experiment design exactly. Participants learned artificial rankings of the eight items, presented as movie popularity rankings based on box office performance. Each participant was assigned a unique ranking sequence and learned the same set of eight non-adjacent pairs through identical few-shot exposure (four presentations per pair across four blocks).

The experiment included two testing phases that employed neural decoding paradigms. In the pre-learning testing phase, participants judged pairs based on real-world size rankings of the objects depicted in the eight images. In the post-learning testing phase, participants judged the same pairs based on the newly learned artificial rankings of these items. This design enabled direct comparison between neural representations of innate knowledge (real-world size) and newly acquired subjective rankings.

Both testing phases employed identical trial structures optimized for MEG analysis. Each trial began with a 400 ms central fixation cross, followed by a 700 ms image presentation. After a maintenance period with central fixation (800–1,000 ms), participants judged whether the current image ranked higher or lower than the previous trial's image. This sequential comparison paradigm allowed examination of neural representations during image processing while maintaining task engagement.

Each testing phase consisted of five blocks containing 81 images each, yielding 80 pairwise judgments per block. To ensure balanced stimulus presentation, we generated 50 randomized sequences of the eight pictures and concatenated them to form the complete testing sequence for each phase.

## MEG data acquisition and preprocessing

MEG recordings were acquired using a whole-head Elekta Neuromag system with 204 planar gradiometers and 102 magnetometers, housed in a magnetically shielded environment. Data were sampled at 1,000 Hz.

Visual stimuli were presented through a rear-projection system with a 32-inch display screen positioned 75 cm from participants. The system delivered stimuli at full HD resolution (1,920 × 1,080 pixels) with a 60 Hz refresh rate. The experiment was implemented in MATLAB using Psychtoolbox, and participant responses were recorded using a dedicated input device.

Data preprocessing was conducted using MNE-Python. Signal Space Separation (Maxfilter) was first applied for noise reduction and motion artifact correction. Subsequent preprocessing steps included bandpass filtering (1–40 Hz), downsampling to 200 Hz, and epoch extraction from −200 to 1,500 ms relative to stimulus onset. After independent component analysis using the FastICA algorithm, Ocular and cardiac components were manually identified and removed.

## Representational similarity analysis

We employed RSA to examine the neural encoding of both real-world size rankings and subject-specific subjective rankings across space and time. To characterize the neural basis of constructed rankings, we employed RSA, which assesses whether neural response patterns reflect hypothesized relational structures by testing whether items that are closer along a given dimension evoke more similar neural activity. This approach allowed us to test whether neural activity patterns reflected shared knowledge versus individualized ranking representations. All analyses were performed using Python *RSAtoolbox* (v.3.0) and MNE-Python.

**Neural RDM construction.** Neural RDMs were computed for each participant at every time point within epochs spanning −200 to 1,500 ms relative to stimulus onset. For each of the eight items, we extracted spatial patterns of neural activity across all 204 gradiometer sensors. Neural dissimilarity between any two items was quantified as 1 minus the Pearson correlation coefficient between their corresponding spatial activity patterns. This yielded participant-specific 8 × 8 neural RDMs at each time point, where each element represented the neural dissimilarity between a pair of items. The resulting RDMs were symmetric matrices with diagonal elements excluded from subsequent regression analyses, since identical items cannot exhibit meaningful dissimilarity with themselves.

**Model RDM construction.** Three model RDMs served as regression predictors. The low-level visual similarity model RDM was identical across all participants and constructed from cosine dissimilarity (1 − cosine similarity) computed across pixels and RGB color channels, capturing bottom-up perceptual differences between stimuli. The size-rank model RDM was also identical across participants, computed from absolute differences in canonical object size rankings (1–8), reflecting shared real-world knowledge about physical object sizes. The subjective-rank model RDMs were uniquely generated for each participant based on their individually reconstructed ranking sequences. This design enabled us to distinguish between neural representations of low-level perception, common semantic knowledge, and personalized subjective orderings.

**Regression analysis procedure.** At each time point, we performed multiple linear regression where each participant's neural RDM served as the dependent variable and all three model RDMs—low-level visual similarity, size-rank, and subjective-rank—served as independent predictors. This regression yielded three β coefficients per participant per time point: $\beta_1$ representing the strength of neural encoding for low-level visual similarity, $\beta_2$ representing the strength of neural encoding for real-world size rankings, and $\beta_3$ representing the strength of neural encoding for subjective rankings.

For low-level visual similarity representations, we quantified neural encoding as the average of $\beta_1$ coefficients across both pre-learning and post-learning phases, reflecting early and stimulus-driven processing that is stable throughout the experiment. For real-world size representations, we quantified neural encoding as the average of $\beta_2$ coefficients across both pre-learning and post-learning phases, reflecting the persistent presence of this innate knowledge throughout the experiment. For subjective ranking representations, we computed the difference between post-learning and pre-learning $\beta_3$ coefficients (post-learning $\beta_3$ minus pre-learning $\beta_3$), isolating neural changes specifically attributable to few-shot learning while using pre-learning activity as a baseline control.

**Statistical analysis.** Statistical significance was assessed through non-parametric cluster-based permutation testing involving 100,000 sign-flips of regression coefficients to generate null distributions for population-level mean beta values. We applied cluster-based permutation procedures with a cluster-defining threshold of $p < 0.05$ to address multiple comparisons across timepoints [52].

**Spatial distribution analysis.** We conducted sensor-level searchlight RSA to examine the spatial distribution of neural representations. For each sensor, we constructed neural RDMs by incorporating activity patterns from the target sensor and its neighbors (mean 8 sensors, range 4–11), averaged across significant time windows identified in whole-brain RSA (165–290 ms for low-level visual similarity; 245–265 ms for size-rank; 685–775 ms for subjective-rank). These sensor-level neural RDMs were regressed against both model RDMs to generate participant-specific spatial β-maps.

**Validation of subject-specific patterns.** To confirm that significant neural representations were genuinely tied to individual participants' unique subjective rankings, we performed a validation analysis. We systematically shuffled subjective-rank assignments across participants while preserving original neural data. For each of 1,000 permutation iterations, we randomly reassigned subjective-rank models among participants and regressed neural signals averaged across the significant time window (685–775 ms) against the shuffled models. This procedure generated a null distribution of $\beta$ values against which we statistically compared the originally observed mean $\beta$ coefficients, confirming whether identified neural representations were participant-specific rather than due to chance alignments.

## Supporting information

**S1 Fig. Pre-registered behavioral results ($N = 40$). (a)** Grand averaged ranking accuracy for learned (light blue; 8 pairs in few-shot learning) and non-learned pairs (dark blue; remaining 20 pairs not directly learned). **(b)** Grand averaged accuracy as function of rank position (serial position effect). **(c)** Grand averaged accuracy as function of ranking distance (distance effect). **(d)** Grand averaged accuracy matrix for all the 24 tested pairs, with row and column denoting corresponding items per pair (Deep-to-light color represents high-to-low accuracy). Eight directly-learned pairs (few-shot learning) were marked by triangles. **(e)** Beta-distribution model fitting results for each pair (brown: high-accuracy, $\alpha > 1$, $\beta < 1$; green; bimodal distribution, $\alpha < 1$, $\beta < 1$. **(f)** Beta-distribution model fitting results for each subject. Each point represents the best-fitting $\alpha$ and $\beta$ parameters for an individual subject. Brown points indicate subjects with bimodal error distributions ($\alpha < 1$, $\beta < 1$); green points indicate subjects exhibiting high-accuracy patterns ($\alpha > 1$, $\beta < 1$). Both $\alpha$ and $\beta$ are significantly smaller than 1 (one-sample $t$ test, excluding 10 high-accuracy participant as pre-registered; $\alpha$, $t(29) = -12.32$, $p < 0.001$; $\beta$, $t(29) = -185.69$, $p < 0.001$). **(g)** Individual-level error consistency (left) and proportion of subjects making consistent error on at least one local pair for different thresholds (0.6 to 1.0). Dots denote individual data. **(h)** Self-consistency was significantly higher than chance (permutation test: $p = 0.002$; observed $= 0.99$). **(i)** Number of subjects for each self-consistency category. The data underlying this Figure can be found in https://osf.io/gya95/.
(TIF)

**S2 Fig. Replication behavioral results ($N = 37$). (a)** Grand averaged ranking accuracy for learned (dark blue; 8 pairs in few-shot learning) and non-learned pairs (light blue; remaining 20 pairs not directly learned). **(b)** Grand averaged accuracy as function of rank position (serial position effect). **(c)** Grand averaged accuracy as function of ranking distance (distance effect). **(d)**

Grand averaged accuracy matrix for all the 24 tested pairs, with row and column denoting corresponding items per pair (Deep-to-light color represents high-to-low accuracy). Eight directly-learned pairs (few-shot learning) were marked by triangles. **(e)** Beta-distribution model fitting results for each pair (brown: high-accuracy, $\alpha > 1$, $\beta < 1$; green; bimodal distribution, $\alpha < 1$, $\beta < 1$. **(f)** Beta-distribution model fitting results for each subject. Each point represents the best-fitting $\alpha$ and $\beta$ parameters for an individual subject. Brown points indicate subjects with bimodal error distributions ($\alpha < 1$, $\beta < 1$); green points indicate subjects exhibiting high-accuracy patterns ($\alpha > 1$, $\beta < 1$). Both $\alpha$ and $\beta$ are significantly smaller than 1 (one-sample $t$ test, excluding 13 high-accuracy participant as pre-registered; $\alpha$, $t(23) = -9.27$, $p < 0.001$; $\beta$, $t(23) = -97.30$, $p < 0.001$). **(g)** Individual-level error consistency (left) and proportion of subjects making consistent error on at least one local pair for different thresholds (0.6 to 1.0). Dots denote individual data. **(h)** Self-consistency was significantly higher than chance (permutation test: $p = 0.002$; observed = 1.00). **(i)** Number of subjects for each self-consistency category. The data underlying this Figure can be found in https://osf.io/gya95/. (TIF)

**S3 Fig. Serial position effect, distance effect for learned and non-learned pair. (a)** Serial position effect. Upper panel (learned): $F(7, 532) = 11.87$, $p < 0.001$; lower panel (non-learned): $F(7, 532) = 11.87$, $p < 0.001$. **(b)** Symbolic distance effect. Upper panel (learned): slope coefficient = 0.02, $t(76) = 4.556$, $p < 0.001$; lower panel (non-learned): slope coefficient = 0.05, $t(76) = 14.141$, $p < 0.001$. **(c)** Transitive distance effect is significant without controlling for rank distance linear mixed-effect model; $F(1, 1463.00) = 6.37$, $p = 0.012$) but not significant after control ($F(1, 1463.00) = 0.17$, $p = 0.68$). Pairs with 0 transitive distance is excluded from this analysis as they are directly learned pairs. The data underlying this Figure can be found in https://osf.io/gya95/. (TIF)

**S4 Fig. BetaQ and Betasort model simulations. (a)** Grand averaged ranking accuracy for learned (dark blue; 8 pairs in few-shot learning) and non-learned pairs (light blue; remaining 20 pairs not directly learned). **(b)** Grand averaged accuracy as function of rank position (serial position effect). **(c)** Grand averaged accuracy as function of ranking distance (distance effect). **(d)** Beta-distribution fitting results for each pair (brown: high-accuracy, $\alpha > 1$, $\beta < 1$; gray; unimodal distribution, $\alpha > 1$, $\beta > 1$). **(e)** Individual-level error consistency based on betaQ (Left) and betasort (Right) models. *Left:* grand averaged error consistency (>0.8, one-sample t test, p > 0.05). dots denote individual data. *Right:* number and proportion of subjects making consistent error on at least one local pair for different thresholds (0.6 to 1.0). The data underlying this Figure can be found in https://osf.io/gya95/. (TIF)

**S5 Fig. Distance-averaging and correct-ranking model simulations. (a)** Grand averaged ranking accuracy for learned (dark blue; 8 pairs in few-shot learning) and non-learned pairs (light blue; remaining 20 pairs not directly learned). **(b)** Grand averaged accuracy as function of rank position (serial position effect). **(c)** Grand averaged accuracy as function of ranking distance (distance effect). **(d)** Beta-distribution fitting results for each pair (brown: high-accuracy, $\alpha > 1$, $\beta < 1$; gray; unimodal distribution, $\alpha > 1$, $\beta > 1$). **(e)** Individual-level error consistency based on Distance-averaging (Left) and correct-ranking (Right) models. *Left:* grand averaged error consistency (>0.8, one-sample $t$ test, $p > 0.05$). dots denote individual data. *Right:* number and proportion of subjects making consistent error on at least one local pair for different thresholds (0.6 to 1.0). The data underlying this Figure can be found in https://osf.io/gya95/. (TIF)

**S6 Fig. Model predicted self-consistency and inter-subject ranking similarity. (a)** Number of subjects by self-Consistency category for BetaQ, Betasort, Distance-Averaging, and Correct-Ranking Models. **(b)** Inter-subject ranking similarity for Human, Q-learning, BetaQ, Betasort, Distance-Averaging, and Correct-Ranking Models. The data underlying this Figure can be found in https://osf.io/gya95/. (TIF)

**S7 Fig. Behavioral results and model fitting of MEG experiment. (a)** Grand averaged ranking accuracy for learned (dark blue; 8 pairs in few-shot learning) and non-learned pairs (light blue; remaining 20 pairs not directly learned). **(b)** Grand averaged accuracy as function of rank position (serial position effect). **(c)** Grand averaged accuracy as function of ranking distance (distance effect). **(d)** Grand averaged accuracy matrix for all the 28 tested pairs, with row and column denoting corresponding items per pair (Deep-to-light color represents high-to-low accuracy). Eight directly-learned pairs (few-shot learning) were marked by triangles. **(e)** Beta-distribution model fitting results for each pair (brown: high-accuracy, $\alpha > 1$, $\beta < 1$; green; bimodal distribution, $\alpha < 1$, $\beta < 1$; gray; unimodal distribution, $\alpha > 1$, $\beta > 1$) for experimental data (left) and Q-learning model (right). Note that simulation results support unimodal distribution that diverges from bimodal experimental findings. **(f)** Individual-level error consistency and proportion of subjects making consistent error on at least one local pair for different thresholds (0.6 to 1.0), for experimental data (left) and Q-learning model (right). Dots denote individual data. **(g)** Consistent local errors for each subject. **(h)** Topographic distributions for low-level visual features using sensor-level search light analysis. The data underlying this Figure can be found in https://osf.io/gya95/.
(TIF)

**S8 Fig. Null relationship between learning sequence similarity and ranking similarity. (a)** Overall correlation across all learning blocks revealed no significant association ($r = -0.015$, $p = 0.423$). **(b)** Separate analyses for each learning block confirmed the null relationship (Block 1: $r = -0.012$, $p = 0.516$; Block 2: $r = 0.006$, $p = 0.758$; Block 3: $r = -0.018$, $p = 0.322$; Block 4: $r = -0.005$, p = 0.798). The data underlying this Figure can be found in https://osf.io/gya95/.
(TIF)

**S9 Fig. Supplemental Between-Group Experiment ($N = 40$). (a)** Beta-distribution model fitting results for each pair (brown: high-accuracy, $\alpha > 1$, $\beta < 1$; green; bimodal distribution, $\alpha < 1$, $\beta < 1$). **(b)** Beta-distribution model fitting results for each subject. Each point represents the best-fitting $\alpha$ and $\beta$ parameters for an individual subject. Brown points indicate subjects with bimodal error distributions ($\alpha < 1$, $\beta < 1$); green points indicate subjects exhibiting high-accuracy patterns ($\alpha > 1$, $\beta < 1$). Both $\alpha$ and $\beta$ are significantly smaller than 1 (one-sample $t$ test, excluding 15 high-accuracy participant as pre-registered; $\alpha$, $t(23) = -10.11$, $p < 0.001$; $\beta$, $t(23) = -142.26$, $p < 0.001$). **(c)** Number of subjects for each self-consistency category. **(d)** Independent samples t test revealed comparable ranking consistency among participants who learned the same stimulus set (within-group) versus different stimulus sets (between-group), $t = 0.02$, $p = 0.99$. This null result suggests that the specific stimulus set used during learning did not systematically influence participants' subsequent ranking strategies. The data underlying this Figure can be found in https://osf.io/gya95/.
(TIF)

**S10 Fig. Comparison of inter-individual variability and within-subject consistency between behavioral and MEG post-learning phase. (a)** Lower inter-subject ranking similarity in MEG post-learning phase compared to behavioral experiment. **(b)** Consistent errors at individual level in MEG post-learning phase compared to behavioral experiment. **(c)** Lower inter-subject ranking similarity in MEG post-learning phase compared to MEG pre-learning phase. **(d)** Within-subject correlations between real-world size rankings and individually reconstructed subjective rankings were essentially zero (mean $r = 0.03$, 95% CI [−0.06, 0.13]). The data underlying this Figure can be found in https://osf.io/gya95/.
(TIF)

**S11 Fig. Deterministic model simulations. (a)** Grand averaged ranking accuracy for learned (dark blue; 8 pairs in few-shot learning) and non-learned pairs (light blue; remaining 20 pairs not directly learned). **(b)** Grand averaged accuracy as function of rank position (serial position effect). **(c)** Grand averaged accuracy as function of ranking distance (distance effect). **(d)** Grand averaged accuracy matrix for all the 28 tested pairs, with row and column denoting corresponding items per pair (deep-to-light color represents high-to-low accuracy). Eight directly-learned pairs (few-shot learning) were marked by triangles. Red dotted box highlights the exemplar pair in **e. (e)** Observed bimodal distribution, i.e., some subjects

consistently made correct inference (3rd < 4th), while others made robust errors (3rd > 4th). **(f)** Beta-distribution fitting results for each pair (brown: high-accuracy, $\alpha > 1$, $\beta < 1$; gray; unimodal distribution, $\alpha > 1$, $\beta > 1$). **(g)** Individual-level error consistency based on Distance-averaging (Left) and correct-ranking (Right) models. *Left:* grand averaged error consistency (>0.8, one-sample $t$ test, $p < 0.05$). dots denote individual data. *Right:* number and proportion of subjects making consistent error on at least one local pair for different thresholds (0.6 to 1.0). **(h)** Local error pattern for each subject. Blank tiles in the lower triangular matrices denote correct pairs (accuracy > 50%). Red tiles denote error pairs (deep-to-light color represents high-to-low error proportion). **(i)** Number of subjects for each self-consistency category. **(j)** Inter-subject ranking similarity for experimental data and Deterministic simulation (two-sample $t$ test, $p < 0.001$). The data underlying this Figure can be found in https://osf.io/gya95/.
(TIF)

## Acknowledgments

We thank the Center for MRI Research at Peking University in Beijing, China, for assistance with data acquisition.

The authors used ChatGPT for language editing and clarity improvements. After using this tool/service, the author(s) reviewed and edited the content as needed and take(s) full responsibility for the content of the published article.

## Author contributions

**Conceptualization:** Muzhi Wang, Huan Luo.

**Data curation:** Dongning Liu, Muzhi Wang.

**Formal analysis:** Dongning Liu, Muzhi Wang.

**Funding acquisition:** Huan Luo.

**Investigation:** Dongning Liu, Muzhi Wang, Huan Luo.

**Methodology:** Dongning Liu, Muzhi Wang.

**Project administration:** Huan Luo.

**Resources:** Huan Luo.

**Software:** Muzhi Wang.

**Supervision:** Huan Luo.

**Validation:** Dongning Liu, Muzhi Wang, Huan Luo.

**Visualization:** Dongning Liu.

**Writing – original draft:** Dongning Liu, Muzhi Wang, Huan Luo.

**Writing – review & editing:** Muzhi Wang, Huan Luo.

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
