## [Editor Report · Decision Letter 0]

10 Oct 2025

Dear Huan,

Thank you for submitting your manuscript entitled "Human brains construct individualized global rankings from identical few-shot local learning" for consideration as a Research Article by PLOS Biology.

Your manuscript has now been evaluated by the PLOS Biology editorial staff and I am writing to let you know that we would like to send your submission out for external peer review.

Once your full submission is complete, your paper will undergo a series of checks in preparation for peer review. After your manuscript has passed the checks it will be sent out for review. To provide the metadata for your submission, please Login to Editorial Manager (https://www.editorialmanager.com/pbiology) within two working days, i.e. by Oct 14 2025 11:59PM.

Kind regards,

Christian

Christian Schnell, PhD

Senior Editor

PLOS Biology

cschnell@plos.org

---

## [Decision Letter · Decision Letter 1]

10 Dec 2025

Dear Huan,

Thank you for your patience while your Short Report "Human brains construct individualized global rankings from identical few-shot local learning" was peer-reviewed at PLOS Biology. It has now been evaluated by the PLOS Biology editors, an Academic Editor with relevant expertise, and by several independent reviewers.

In light of the reviews, which you will find at the end of this email, we would like to invite you to revise the work to thoroughly address the reviewers' reports.

As you will see below, the reviewers write that the study provides potentially important insights. However, they all suggest a number of additional analyses, clarifications and textual revisions that are necessary to strengthen the support for the claims.

Given the extent of revision needed, we cannot make a decision about publication until we have seen the revised manuscript and your response to the reviewers' comments. Your revised manuscript is likely to be sent for further evaluation by all or a subset of the reviewers.

**IMPORTANT - SUBMITTING YOUR REVISION**

*Re-submission Checklist*

*Published Peer Review*

*PLOS Data Policy*

*Blot and Gel Data Policy*

Sincerely,

Christian

Christian Schnell, PhD

Senior Editor

PLOS Biology

cschnell@plos.org

REVIEWS:

Reviewer #1: This study provides potentially valuable insights into the constructive and generative nature of human relational learning under sparse data conditions. The use of a preregistered few-shot learning paradigm combined with behavioral experiments and MEG recordings is a novel methodological approach, allowing for both behavioral and neural evidence to support the findings. The study challenges existing models of relational learning by demonstrating that individuals construct idiosyncratic global rankings even when exposed to identical learning inputs, highlighting the role of individual inductive biases and memory constraints.

My general concern is that, throughout the paper, the rationale for the experimental design, the description of results, and the interpretation of those results need to be described in much greater detail with much more attention to the logical flow connecting the design, results and conclusions. I have two specific sets of concerns:

First, the authors have invented a novel way to train ordinal relations, but it isn't clear why they did this or what they expected to find. The conventional approach is to train adjacent pairs. Given a list of n items, if all n-1 adjacent pairs are trained, then the participant has complete information about the list order. However, in the current study, the authors train non-adjacent pairs but also provide a cue signaling the distance between the items in each pair. The authors do not fully explain the rationale for their method, but they indicate that they believe that participants (P's) received only partial information about the list order, leading them to make persistent idiosyncratic errors. I do not believe this is a correct description of the task or the P's behavior. I believe that, by providing 8 non-adjacent pairs plus relative distance, the authors' method presents information that uniquely specifies the order of the 8 list items. Hence, their method provides the same information as the conventional adjacent-pair method, just in a less direct manner. I do not believe that the authors can claim that their method presents "partial pairs" if it in fact presents the same information as a complete set of adjacent pairs. Any idiosyncrasy in behavioral responses could be due to P's inability to extract the information available rather than their filling in missing information.

An easy way to make use of all the information provided is to average the signed distance for each pair to derive a metric for each item. For example, the tire appears twice, with respective distances of +3 and +7. It can therefore be assigned an average distance of +5. The sandwich also appears twice with signed distances of -5 and -7, giving an average of -6. Applying this logic across the entire training set yields a set of average distances that order the objects in a way that is close to their true ranks. If participants employ this strategy, they should make the most errors on items E and F. Looking at the data in Fig. 2, it appears that this may be the case.

It should be noted that P's are also given the following ordinal relations:

C>H, F>G, D>G, B>F, C>E, B>E, A>D and A>H. From these, one can infer that B>G, A>G. Therefore, one might expect higher accuracy on these pairs than on other test pairs.

Overall, it is critical to clearly communicate the rationale for the experimental design and provide a correct analysis of whether there is enough information for P's to learn the entire list order if they were able to use all the information presented.

The second set of concerns has to do with the MEG results. A simpler way to describe the MEG results is that they might reflect the difference between a visual comparison and an abstract comparison. The interpretation that MEG reflects an internal ranking needs to be explained more clearly. Why was representational dissimilarity used? How does it produce a representation of rank? It is not clear exactly how abstract rank was decoded. To test the decoder, it should work on multiple lists for the same participant. There is also a potential performance confound as overall performance was not equal between the different ordering tasks.

Reviewer #2: Liu et al. presents a study on a transitive inference paradigm with behavioral and MEG measures. The central hypothesis is that few-shot learning from limited information during transitive inference is a constructive process driven by idiosyncratic factors across individuals. Hence, even when providing the same learning pairs during inference, the authors hypothesize that different individuals will have distinct ranking errors that are consistent across repeated testing of the same pairs, reflecting self-consistent global ranking that is incorrect in individualized ways across subjects. The study has several advantages, including using a large sample size for the behavioral study, comparing against null hypothesis based on computational modeling, and using MEG data with RSA to investigate neural correlates. While the overall findings are well presented, several questions impede a clearer contribution to the literature in this study. Hence, I recommend revisions to address these key points before being evaluated for publication.

MAJOR 1: The main focus of the study is about showing how individuals exhibit idiosyncratic local pair errors under few-shot learning despite the same learning pairs. It is perhaps not very surprising that under a difficult transitive inference setup, individuals will make mistakes, and those mistakes may not agree between individuals. One thread that the authors touched on is how the current models and theories of relational learning overlook this individualized difference. But it is not clear to readers why do the normative models of transitive inference prohibit individualized learning outcomes. Is the main point just that the computational models do not have stochasticity to account for the observed global ranking errors? Perhaps a better explanation of the implication of having or not having individualized learning in the extant models can better contextualize the importance of the current findings. In addition, how do the current results agree with previous studies showing the success of transitive inference models in explaining participants behaviors? If such strong inter-individual variability also existed in previous studies, one would have expected poor model fitting. Figure 3 e. shows that even though subjects make local errors, the overall performance is quite good and there is certainly a correlation with the ground-truth global ranking across subjects (darker color to the right of bars). The authors discussed differences in the paradigm from standard transitive inference, so how much of the observed idiosyncratic ranking just due to a harder paradigm where subjects? Or perhaps the same level of variability has always existed but just did not impact the general agreement between model-based relational learning and subject performance? Some level of variability must be expected in such task, just like response time is not identical across subjects. So how should readers appreciate the effect size of the variability reported here? These should be better described and discussed - otherwise it is unclear why the current results are meaningful to the literature on relational learning beyond just "noise".

MAJOR 2: One aspect of the learning phase that concerns me is the order of learning pairs. The methods section stated that the ranking sequence of each participant was randomly shuffled in the learning phase. So even though the exact eight pairs used in few-shot learning is random, but the order with which they are presented is different across subjects. Then a major question is how much the ordering of the learning pair presentation influenced the idiosyncratic learning across subjects. Had subjects been given the same ordering of the same learning pairs, making it truly identical, would subjects show more consistency in constructed global ranking? This seems to be a major confound given the central claim is about individualized global ranking and learning outcomes.

MAJOR 3: The MEG task differs from the behavioral-only paradigm in an important way: the pre-learning biased subjects to evaluate the images on a common and accessible semantic knowledge dimension: size. Then the few-shot learning requires subjects to unlearn and overcome this pre-learning phase bias in order to learn the actual arbitrary global ranking among the objects. A comparison of the level of inter-individual variability as well as within-subject consistency in errors across the MEG and behavioral tasks will be helpful. Such result can help readers appreciate the effect size of the reported variability results on the basis of how much a pre-existing semantic knowledge influences the relational learning. In real life, people often have variable semantic knowledge and intrinsic feature values, so how much is "identical local learning" really relevant in ecological settings is unclear.

MAJOR 4: The beta weights reported in Figure 4 should be better unpacked. Model RDM 2 should not have any meaningful beta weights during pre-learning, since the subjects haven't been exposed to the arbitrary ranking yet, it doesn't make sense that it should explain any neural RDM pattern. Because of that, it is strange to subtract post-learning from pre-learning for the subject-specific rank - why not just use the post-learning alone? Also, g and i have similar scales. But one would have expected a summation of two beta weights to be larger than the difference between two beta weights. Therefore, these results should be better presented/explained.

MINOR 1: The authors keep using the term "non-adjacent" pairs throughout the text. But in Figure 1 a and b, isn't the violin and apple pair adjacent rather than "non-adjacent"?

Reviewer #3: The current study aimed to better understand how humans form rankings of items. For this, they used a few shot learning approach of an 8-item ranking. Participants saw four blocks of eight pairs from the possible 28 pairs; each pair was accompanied with a bar to depict the item value allowing to learn the relative value of each pair with an individual offset from zero without informational value. During testing, each possible pair was shown and participants decided which was worth more. This was done ten times in blocks. In an additional experiment, participants performed a similar task in the MEG. Here the testing occurred in a long sequence and a judgement was made for each item and the preceding item. The behavioral data mainly showed that participants made individual mistakes, i.e., there was no common pattern in the errors. The MEG data mainly showed that representations of the learned ranking can be observed after roughly 700 ms. In general, the study is performed with high methodological rigor, although I do not find the conclusions to be overly novel. When reading, I had some specific concerns.

1. I am unsure that I understood the modeling correctly. The authors use Q-learning modeling and flavors thereof (they give no rational for the specific models). Learning was without feedback or choice (observing value pairs) and testing was without feedback. It is unclear to me how the values are learned by the model, if no "prediction error" occurs. Can the authors explain more clearly, how values were updated? The paper states that observed absolute values were non-informative and only the difference matters. It would seem that value-updating can become additionally distorted by this. Maybe the authors can provide a source, where the rationale of this type of modeling is validated. Additionally, I would like to know if the authors would argue that value learning by explicitly showing value pairs is learned by reinforcement learning mechanisms.

2. In general, the authors need to be more clear that only the behavioural study was preregistered. I found this quite misleading. In addition, I found the preregistration quite short and hard to understand. Hence, it was challenging to understand where authors deviated from the preregistered analysis plan. I would ask the authors to make this explicit in the manuscript. Additionally, the preregistration states sampling of 40 participants, but nearly twice this amount was recruited! Although this is acknowledged in the manuscript, it is only said that this was done to further validate the findings, but it is not said how authors deal with changes in the stopping rule and resulting alpha inflation!

3. When reading the introduction, it was hard to understand the study's rationale and how it advances previous work. It seems intuitively plausible that participants would form idiosyncratic rankings based off their learning experience even if the learning experience is kept constant. This is because the brain's ability to encode information fluctuates constantly, e.g., at the millisecond scale due to ongoing brain oscillations and at the minute scale due to attention allocation (not to mention circadian rhythms etc.). This will automatically lead to individual participants encoding and consolidating some trials and failing to encode and consolidate others with no correlation between subjects. The finding of a bimodal distribution in responses thus seems to result from this well-known fact. I understand that the participants had 4 blocks of learning, but essentially getting rid of this effect would require learning to 100, which defeats the purpose. Additionally, this bimodality emerges mainly for distant unlearned pairs, which are the more difficult inferences to make and here participants seem to be showing a consistent error pattern. Again not surprising or new. Unless I am missing something, this "effect" is common knowledge in the field.

4. Was the serial position effect and symbolic distance effect (Figure 1 e + f) calculated including the learned pairs or only the unlearned pairs?

5. The MEG RSA-results are reported only after averaging or subtracting. It would be important to judge the raw betas before and after the learning phase. Please add time series (like 4 g + i) with pre- and post-learning betas to Figure 4.

6. In the MEG experiment participants were excluded for poor learning performance. This needs to be explained in more detail. How did the results change, if these participants are included?

Typos: l85: „consistent global orders that deviates from", l385 "summary, out study highlights"

---

## [Decision Letter · Decision Letter 2]

17 Mar 2026

Dear Huan,

Thank you for your patience while we considered your revised manuscript "Humans construct idiosyncratic, self-consistent global rankings from few-shot local evidence" for consideration as a Short Reports at PLOS Biology. Your revised study has now been evaluated by the PLOS Biology editors, the Academic Editor and the original reviewers.

In light of the reviews, which you will find at the end of this email, we are pleased to offer you the opportunity to address the remaining points from Reviewer 3 in a revision that we anticipate should not take you very long. We will then assess your revised manuscript and your response to the reviewers' comments with our Academic Editor aiming to avoid further rounds of peer-review, although we might need to consult with the reviewers, depending on the nature of the revisions.

**IMPORTANT - SUBMITTING YOUR REVISION**

*Resubmission Checklist*

*Published Peer Review*

*PLOS Data Policy*

Sincerely,

Christian

Christian Schnell, PhD

Senior Editor

PLOS Biology

cschnell@plos.org

REVIEWS:

Reviewer #1: The authors have addressed all of my original concerns. I have no further comments.

Reviewer #2 (Mingjian He has signed his report): All my comments have been addressed.

Reviewer #3: Thank you for the extensive revisions. The ms is much improved! Some remarks remain, however.

1. I would like to thank the authors for their clear explanation of the process. I did understand now much better the underlying ideas. However, I remain doubtful. This is because in absence of choice at learning, how should the fitted values functions differ between different participants? This would be crucial for idiosyncratic errors. Maybe I am missing something, but it seems logical that the models do not have idiosyncrasies if they all had the exact same learning experience. I mean, if they do, I would be eager to see these data. Humans on the other hand will have fluctuating attention and thus all differ between each in the information they have access to leading to idiosyncratic errors (see my comment 3).

2. Thank you for clarifying.

3. I think this is crucial. The novelty of the finding is based on the authors' claiming it not being modelled by the computational models, but these models do not include things like attention. Maybe a fix would be to extend the model to include attention effects, e.g., by fitting the Q-learning and allowing missing trials (possibly even as a fit-able parameter). Possibly this would lead to models being idiosyncratic, too? In any case a positive model would strongly convince me that the claim the authors are making is true.

4. Wow. Great to see � It might be worth calculating stats on the slope of the symbolic distance effect. It seems larger in the non-learned pairs, which seems logical, but nonetheless is an interesting finding!

5. Thank you for providing pre- and post-data. Judging from these curves, it does not seem that there is a distinct difference between the two curves and significance emerges due to the subtraction procedure. I would urge the authors to clearly point this out in the limitations section and to inform readers that additional research is required. Out of curiosity, does the non-significant cluster permutation test for the post subjective ranking yield a cluster at the same time point? This would strengthen the authors' claim!

6. Add the info to the ms that the effects are non-significant, if all participants are included. As the authors point out, this is in line with expectations.

---

## [Editor Report · Decision Letter 3]

25 Mar 2026

Dear Huan,

Thank you for your patience while we considered your revised manuscript "Humans construct idiosyncratic, self-consistent global rankings from few-shot local evidence" for publication as a Research Article at PLOS Biology. This revised version of your manuscript has been evaluated by the PLOS Biology editors and the Academic Editor.

Based on our Academic Editor's assessment of your revision, we are likely to accept this manuscript for publication, provided you satisfactorily address the following data and other policy-related requests:

* We would like to suggest a different title to improve its accessibility for our broad audience:

Human brains construct individualized global rankings even when exposed to identical learning input

* Please add the links to the funding agencies in the Financial Disclosure statement in the manuscript details.

* Please include information in the Methods section whether the study has been conducted according to the principles expressed in the Declaration of Helsinki.

* DATA POLICY:

Regardless of the method selected, please ensure that you provide the individual numerical values that underlie the summary data displayed in the following figure panels as they are essential for readers to assess your analysis and to reproduce it: 1EFHI, 2EG, 3BD, 4CE, S1ABGI, S2ABGI, S3A, S4ABE, S5ABE, S6AB, S7ABF, S8AB, S9D, S10ABC and S11ABIJ.

* CODE POLICY

Per journal policy, if you have generated any custom code during the course of this investigation, please make it available without restrictions. Please ensure that the code is sufficiently well documented and reusable, and that your Data Statement in the Editorial Manager submission system accurately describes where your code can be found. More information on our Code Policy, what and how to share can be found here: https://journals.plos.org/plosbiology/s/code-availability

We expect to receive your revised manuscript within two weeks.

*Published Peer Review History*

*Press*

Sincerely,

Christian

Christian Schnell, PhD

Senior Editor

cschnell@plos.org

PLOS Biology

---

## [Editor Report · Decision Letter 4]

27 Mar 2026

Dear Huan,

Thank you for submitting your revised manuscript "Human brains construct individualized global rankings from identical few-shot learning input" for publication as a Research Article at PLOS Biology.

I have now gone through the changes and noted that two items remain open:

1) Please add the links to the funding agencies in the Financial Disclosure statement in the manuscript details.

2) DATA POLICY:

Regardless of the method selected, please ensure that you provide the individual numerical values that underlie the summary data displayed in the following figure panels as they are essential for readers to assess your analysis and to reproduce it: 1EFHI, 2EG, 3BD, 4CE, S1ABGI, S2ABGI, S3A, S4ABE, S5ABE, S6AB, S7ABF, S8AB, S9D, S10ABC and S11ABIJ.

I have checked the submission files and the OSF repository but was unable to locate the source data file.

We expect to receive your revised manuscript within two weeks.

*Published Peer Review History*

*Press*

Sincerely,

Christian

Christian Schnell, PhD

Senior Editor

cschnell@plos.org

PLOS Biology

---

## [Editor Report · Decision Letter 5]

30 Mar 2026

Dear Huan,

Thank you for the submission of your revised Research Article "Human brains construct individualized global rankings from identical few-shot learning input" for publication in PLOS Biology. On behalf of my colleagues and the Academic Editor, Vincent Ferrera, I am pleased to say that we can in principle accept your manuscript for publication, provided you address any remaining formatting and reporting issues. These will be detailed in an email you should receive within 2-3 business days from our colleagues in the journal operations team; no action is required from you until then. Please note that we will not be able to formally accept your manuscript and schedule it for publication until you have completed any requested changes.

While you attend to the requests to come, please also make sure to cite the location of the source data clearly in all relevant main and supplementary Figure legends, e.g. “The data underlying this Figure can be found in https://osf.io/gya95/”

PRESS

We frequently collaborate with press offices. If your institution or institutions have a press office, please notify them about your upcoming paper at this point, to enable them to help maximize its impact. If the press office is planning to promote your findings, we would be grateful if they could coordinate with biologypress@plos.org. If you have previously opted in to the early version process, we ask that you notify us immediately of any press plans so that we may opt out on your behalf.

Sincerely,

Christian

Christian Schnell, PhD

Senior Editor

PLOS Biology

cschnell@plos.org